



# Experimental analysis of radially resolved dynamic inflow effects due to pitch steps

Frederik Berger[1], David Onnen[1], J. Gerard Schepers[2,3], and Martin Kühn[1]

[1]ForWind - Center for Wind Energy Research, University of Oldenburg, Institute of Physics, Küpkersweg 70, 26127 Oldenburg, Germany
[2]TNO Energy Transition, Petten,1755 LE, The Netherlands
[3]Hanze University of Applied Sciences, Groningen, 9747 AS, The Netherlands

**Correspondence:** Frederik Berger (frederik.berger@uol.de)

**Abstract.** The dynamic inflow effect denotes the unsteady aerodynamic response to fast changes in rotor loading due to a gradual adaption of the wake. This does lead to load overshoots. The objective of the paper was to increase the understanding of that effect based on pitch step experiments on a 1.8 m diameter model wind turbine, which we performed in the large open jet wind tunnel of ForWind - University of Oldenburg. We measured the flow in the rotor plane with a 2D Laser Doppler Anemometer and were able to extract the dynamic wake induction factor transients in axial and tangential direction. Further, integral load measurements with strain gauges and hot wire measurements in the near and close far wake were performed. Our results show a clear gradual decay of the axial induction factors after a pitch step, giving the first direct experimental evidence of dynamic inflow due to pitch steps. We fitted two engineering models to the induction factor transients to further investigate the relevant time constants of the dynamic inflow process. We discussed the radial dependency of the axial induction time constants as well as the dependency on the pitch direction. We confirmed that the nature of the dynamic inflow decay is better described by two rather than only one time constant. The dynamic changes in wake radius were connected to the radial dependency of the axial induction transients. In conclusion, the comparative discussion of inductions, wake deployment and loads facilitated the improved physical understanding of the dynamic inflow process for wind turbines. Furthermore, these measurements provide a new detailed validation case for dynamic inflow models and other types of simulations.

## 1 Introduction

Dynamic inflow describes the unsteady response of loads to fast changes in rotor loading, for example, due to fast pitching of the rotor blades or gusts. This unsteady aerodynamic effect leads to load overshoots due to the inertia of the global flow field, as the axial wake induction in the rotor plane cannot change instantaneously but only gradually to a new equilibrium flow field.

In addition to the direct impact on the dynamic loading, van Engelen and Hooft (2004) emphasise the need to model these dynamic inflow effects for the pitch controller design to enhance the stability and thus reduce unnecessary fatigue loads with optimised pitching transients, especially near rated operation. The dynamic wake behaviour due to load changes is intrinsically considered in higher fidelity approaches as Computational Fluid Dynamics (CFD) and Free Vortex Wake Method (FVWM) simulations, thus modelling the dynamic inflow effect. However, engineering models are required to mimic this effect in Blade



Element Momentum (BEM) theory, which is commonly applied for aeroelastic simulations for the design and certification of
wind turbines. Well-tuned engineering models help to avoid too conservative predictions of fatigue loads.

First extensive studies in the 1990s within the Joule I and II projects on the development of dynamic inflow models for
wind turbines are described in Snel and Schepers (1995) and Schepers and Snel (1995). There the free field measurements of
out-of-plane blade root bending moment and rotor shaft torque for pitch steps on the 2 MW Tjæreborg wind turbine, described
in Øye (1991), are used for validation.

Later Schepers (2007) employed a one time constant model to analyse force transients. These were derived from pressure
sensor arrays at five radial stations, after the pitch steps of the NREL phase VI turbine with 10 m diameter (see Hand et al.
(2001)) in the NASA Ames wind tunnel. Forces for the pitch step to low load adapted faster to the new equilibrium than for
the step to high load. They could not experimentally validate the strong radial dependency of the time constant, which they
expected from cylindrical wake models. Sørensen and Madsen (2006) also investigated the same experiment and compared
it to unsteady Reynolds Averaged Navier Stokes (uRANS) CFD simulations. They suggest using a two time constant model
to capture the dynamic inflow effect on the forces. The fast time constant represents the near wake dynamics and decreases
with radius and the slow time constant represents the far wake dynamics. Later Pirrung and Madsen (2018) investigated this
experiment and uRANS CFD simulation again and compared them to a cylindrical wake model. Based on varying the wake
length in the cylindrical wake model they affirm, that two different time constants best describe the dynamic inflow effect.

In the MEXICO project, pitch steps were performed on a 4.5 m diameter model wind turbine, featuring pressure distribution
measurements at five radial stations, as well as in high and mid-fidelity simulations (Boorsma et al. (2018)). They found that
unsteady aerodynamic effects on the blade chord level, namely the Theodorsen effect, reduce the load overshoot. This effect
can be modelled as a time lag on the angle of attack in the order of the ratio of relative wind to respective chord length. In
contrast, the typical dynamic inflow time constant is in the order of the ratio of radius to free wind velocity and two orders of
magnitude higher.

Yu et al. (2017) used an actuator disk with variable blockage in a wind tunnel to study the wake evolution after a change
in thrust. We also performed a preliminary pitch step experiment, focusing on the integral turbine loads (see Berger and Kühn
(2018)). The relevance of improved modelling of the dynamic inflow effect can be seen in the recent development of new
dynamic inflow models by Yu et al. (2019), Madsen et al. (2020) and Ferreira et al. (2021).

Schepers and Schreck (2018) emphasise on the value of experimental investigations of aerodynamic effects and also specif-
ically the dynamic inflow effect to further improve and validate models. Higher fidelity simulations depend on calibration and
thus cannot solely fill this gap. Further, Schepers and Schreck (2018) outline the importance of radius resolved aerodynamic
measurements over integrated blade and rotor loads. No experimental investigation is available until now, where the wake
induction is directly probed at various blade radii in the rotor plane.

The objective of this paper is to get deeper insights into the dynamic inflow effect for wind turbines due to pitch steps.
The main novelty in this work is the dynamic induction measurement. We investigated the radial dependency and differences
between the pitch directions using time constant analysis. Furthermore, the behaviour of the flow in the near and close far
wake and integral loads is used to compare the differences between the pitch directions. These different measurements are





contemplated together to allow for new insights into the dynamic inflow effect, test presumptions and validate findings of prior

works.

## 2 Methods

Here in Sect. 2.1 the experiment is introduced. In Sect. 2.2 time constant models and the fitting approach are outlined. Lastly, in Sect. 2.3 the method for the load reconstruction based on the obtained wake inductions is outlined.

### 2.1 Experiment

In this subsection, all relevant information on the experiment is introduced, consisting of the setup, experimental matrix, wake induction derivation from measurements, ensemble averaging approach, as well as correction models.

#### 2.1.1 Setup

The experiments were performed in the large wind tunnel at ForWind - University of Oldenburg. It is a Göttingen type wind tunnel that can be operated in an open jet and a closed test section configuration. The test section length measures $30\,\mathrm{m}$ and

the rectangular wind tunnel nozzle $3\,\mathrm{m}$ by $3\,\mathrm{m}$, as shown in Fig. 1 a. Wind velocities in the open jet configuration reach up to $32\,\mathrm{ms}^{-1}$. Kröger et al. (2018) provide detailed information about the wind tunnel and the optional active grid. No active grid was used in the measurement and the turbulence intensity of the inflow was in the order of $0.3\,\%$.

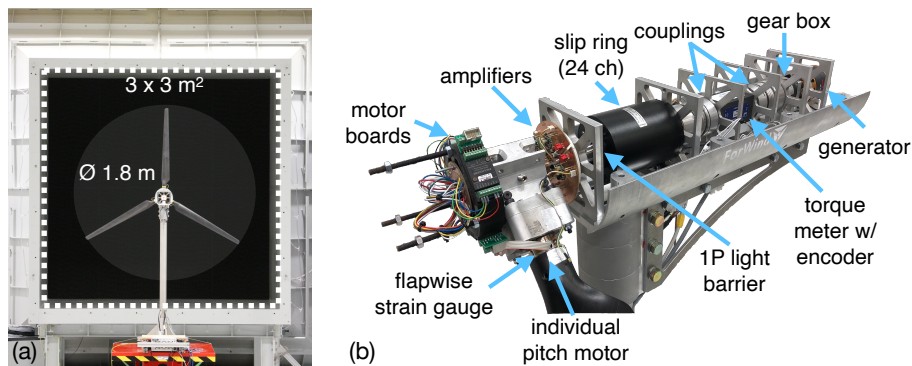

**Figure 1.** (a) Wind tunnel nozzle and MoWiTO 1.8 with main dimension. (b) MoWiTO 1.8 with open nacelle.

The utilised Model Wind Turbine Oldenburg has a diameter of $1.8\,\mathrm{m}$ (MoWiTO 1.8). The machine is an aerodynamically scaled version of the NREL 5 MW generic turbine (see Jonkman et al. (2009)) with a length scaling factor of $n_{length} = \frac{1}{70}$.

The scaling approach was to maintain the design tip speed ratio, thrust and power characteristic, as well as non-dimensional lift and thus axial induction distribution for the operational range. Low Reynolds number airfoils were used. The rotor blades have a stiff carbon fibre-based structure with the first eigenfrequency of $32\,\mathrm{Hz}$ and estimated maximum tip deflections of $0.01\,\mathrm{m}$ The scaling approach and turbine are described in detail in Berger et al. (2018). The blockage ratio of the turbine in the used open





jet configuration wind tunnel is $0.28$, however, Ryi et al. (2015) showed that blockage effects are negligible for such an open
jet configuration.

The MoWiTO 1.8 nacelle is shown in Fig 1 b. The turbine features individual pitch motors, which are mounted in the root of each blade. The pitch motors are small DC motors with a three stage planetary gearbox with a gear ratio of 159:1 and integrated encoders. They are mounted pre-tensioned with springs to counteract gear backlashes and thus allow setting precise pitch angles. Pitching speeds up to $100°\,\mathrm{s}^{-1}$ can be achieved. The main shaft is supported by two roller bearings and connected
by a coupling to a torque meter with an integrated encoder and through another coupling and one stage planetary gearbox to the generator. Flapwise blade root bending moments for each blade are measured by a full Wheatstone bridge strain gauge configuration on the metal adapter, the carbon blades are glued on. The power supply for the amplifiers and motor boards in the hub and their communication with the control hardware is channelled through a slipring from the rotating hub to the stationary nacelle. Further, the thrust of the turbine is derived from a strain gauge measurement of the tower foot bending in fore-aft
direction, as outlined in Sect. 2.1.5. A National Instruments Compact Rio is used for control and data acquisition. Analog data (e.g. strain gauges, external hot wires) is sampled at 5 kHz and the control loop and pitch motor communication run at 100 Hz.

The setup of the MoWiTO 1.8 in the wind tunnel is sketched in Fig. 2. The wind speed is obtained by the measured pressure drop in the wind tunnel nozzle. The turbine is positioned 2.6 diameter ($D$) behind the wind tunnel nozzle, so that the induction zone of the turbine is not influenced (see Medici et al. (2011)).

Integral loads of flapwise blade root bending moment ($M_{\mathrm{flap}}$), rotor thrust ($F_{\mathrm{thrust}}$) and rotor torque ($M_{\mathrm{aero}}$) are obtained based on strain gauge measurements shown in blue in the sketch.

Hot wire measurements in the near wake (up to $1\,D$ according to Vermeer et al. (2003)) and beginning far wake (more than $1\,D$) are performed at hub height. In flow direction (x-axis), seven distances ranging from $0.5\,D$ to $2\,D$ behind the turbine are considered in steps of $0.25\,D$ and shown in red in the sketch. In radial direction (negative y-axis), values between the rotor axis
at 0 radii (R) to $1.4\,R$ in steps of $0.2\,R$ are considered. This adds up to 56 measurement positions.

In the rotor plane, Laser Doppler Anemometer (LDA) measurements are performed with a 2D system by TSI Inc.. A beam expander with a focus length of $2.1\,\mathrm{m}$ is used to not disturb the flow. Both lasers have a maximum power of $1\,\mathrm{W}$. The LDA probe is mounted on a three axes traverse system and can be motor-driven by $1.5\,\mathrm{m}$ in each direction. Measurement points are in the rotor plane at hub height. They are positioned radially (negative y-axis) from $0.25\,R$ to $0.95\,R$ with steps of $0.1\,R$
between $0.3\,R$ and $0.9\,R$ and the smaller extra steps at the edges of the range. The LDA measurement is indicated in green in the sketch and the colour coding for the three signal types, strain gauge, hot wire and LDA, is maintained for all plots over time in this paper.

### 2.1.2 Experimental matrix

The turbine is operated at a rotational speed of 480 rpm and wind velocity of $6.1\,\mathrm{ms}^{-1}$. This corresponds to a tip speed of
$45\,\mathrm{ms}^{-1}$ and a tip speed ratio of 7.4. Chord based Reynolds numbers range from a minimum value of $60 \cdot 10^{3}$ at the first airfoil at $0.2\,R$ to values between $100 \cdot 10^{3}$ and $120 \cdot 10^{3}$ from $0.5\,R$ to the tip. Time constants in dynamic inflow models are related to a reference time constant $\tau_{ref} = R/u_0$ (see Schepers (2012)). This amounts to $0.15\,\mathrm{s}$ here.



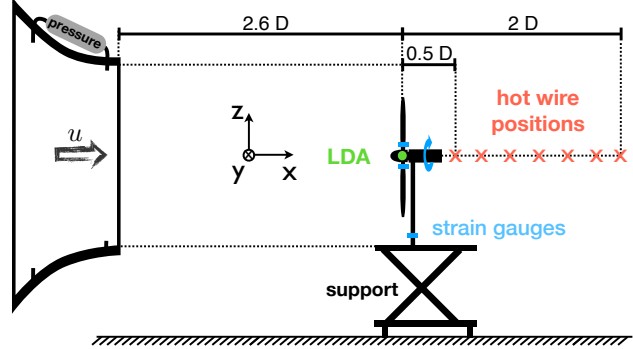

**Figure 2.** Sketch of the setup in the wind tunnel.

The rotor blades are collectively pitched by $5.9°$ within $0.070\,\mathrm{s}$, corresponding to about half a rotor revolution and half the reference time. The pitch step is from a low rotor load at a thrust coefficient $C_T = 0.48$ to a high load at $C_T = 0.90$ and vice versa, based on the strain gauge derived thrust. This corresponds to rotor effective inductions of $a_{eff} = 0.14$, respectively $a_{eff} = 0.34$, based on the momentum theory relation ($C_T = 4a(1-a)$).

The representative encoder reading of one pitch motor is plotted in Fig. 3. Between the pitch steps, there are $3\,\mathrm{s}$ (24 revolutions) to allow for the flow to reach an equilibrium again. There is a slight overshoot of the pitch angle for both pitch directions by one encoder step ($0.18°$), which due to the small value has no noticeable effect on our investigation.

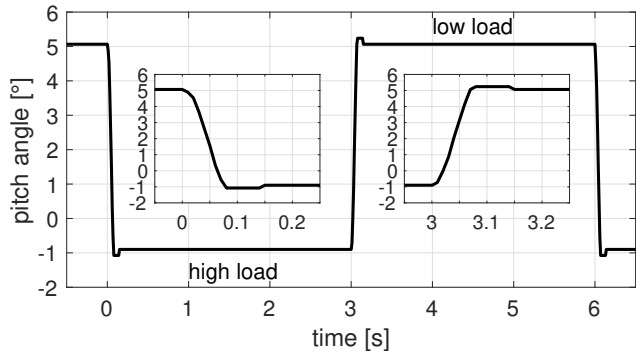

**Figure 3.** Pitch motor encoder signal for a pitch step to high load followed by a step to low load with additional zoomed in views of the actual pitch steps.

For the LDA measurements, 100 pitch steps were performed for each radial position and pitch direction at a typical sampling frequency of $600\,\mathrm{Hz}$. This sampling frequency is just an order of magnitude since it depends on many parameters, especially the seeding of tiny oil droplets in the wind tunnel and thus varies constantly. Load measurements are taken during the LDA measurements. Therefore load signals for 900 pitch steps are available. The hot wire measurements have been performed separately. A wake rake consisting of four hot wires was used to measure at the described 56 positions in the wake. Thus the





experiment, consisting of 25 pitch steps, had to be performed $56/4 = 14$ times and the wake rake was moved between those measurements.

### 2.1.3   Wake induction measurement by 2D-LDA

The wake induction is derived from the LDA measurements by a method introduced by Herráez et al. (2018) for steady operation. The method uses the local velocity in the rotor plane, free of the influence of the bound circulation. This velocity
is obtained by probing in the bisectrix of two rotor blades for axial and uniform inflow. In the bisectrix, the blade induction is counterbalanced and thus cancelled out. This method is less suited for the root and tip region of the blade, as the effect of trailed vorticity is not caught. In Fig. 4 a the MoWiTO turbine is shown with the LDA laser beams and the probed axial ($u_{ax}$) and tangential ($u_{ta}$) velocity components at a specific radius. Alongside in Fig. 4 b, the concept of the counterbalancing of the bound circulation of the evenly loaded blades is sketched. The tower does disturb the axial symmetry, however, based on
an estimation of the tower effect with a dipole model as in Schepers (2012) the tower effect at the measurement positions is considered negligible.

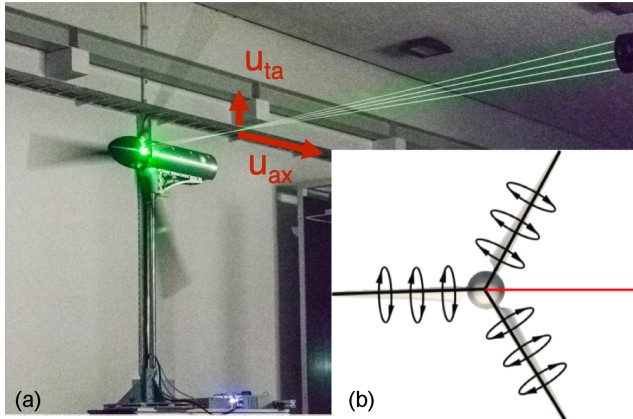

**Figure 4.** (a) Rotating MoWiTO 1.8 with 2D-LDA probing axial and tangential velocity components in the bisectrix of two blades. (b) Scheme of principle of counterbalancing bound circulation of the evenly loaded blades in the bisectrix, indicated in red (adapted from Herráez et al. (2018)).

To obtain the values in the bisectrix, we synchronised the LDA system with the MoWiTO data acquisition system. Measurements at constant load are plotted for one position of the axial and tangential probe over the azimuth angle $\phi_1$ of the turbine in Fig. 5. The bisectrix values that are in a threshold of $\pm 3°$ are marked in red. We identified these threshold values to give a
good compromise between data points and data quality. Note that the axial probe does look like the theoretical derivation of the signal in reference Herráez et al. (2018). For the tangential probe, data is missing around $-1.3\,\mathrm{m\,s}^{-1}$ and also at $2.3\,\mathrm{m\,s}^{-1}$ for the axial probe, which is due to the beta status of the LDA system at that point in time. We were aware of this bug and it has no influence on the presented measurements.



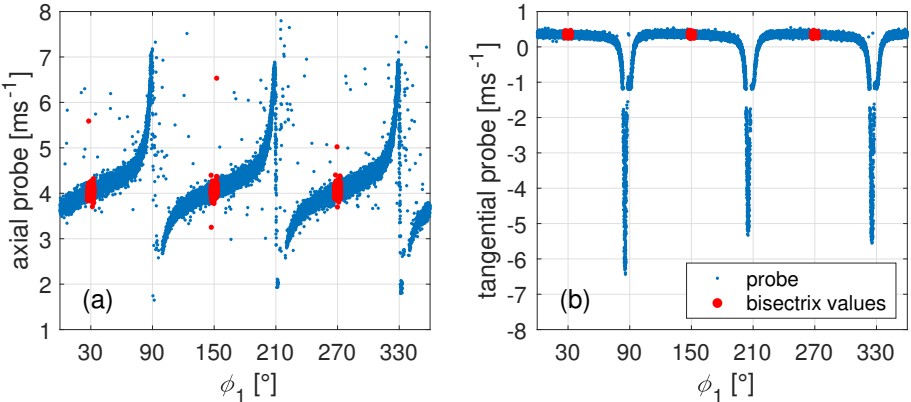

**Figure 5.** (a) Measurements of axial probe for high load case at a radius of $0.7\,R$ for 400 revolutions over azimuth angle $\phi_1$ with marked data within the bisectrix threshold. $\phi_1 = 0°$ relates to the 12 o'clock position of blade 1. (b) Analogously for the tangential probe.

Based on these measured axial and tangential velocities, the undisturbed inflow velocity $u_0$ and the angular velocity $\Omega$, the axial ($a$) and tangential ($a'$) wake induction factors are defined by Eq. (1) and Eq. (2). With the geometrical angle of the rotor segment ($\theta$), consisting of twist and pitch, the angle of attack $\alpha$ can be calculated by Eq. (3).

$$a = 1 - \frac{u_{ax}}{u_0} \tag{1}$$

$$a' = \frac{u_{ta}}{\Omega r} \tag{2}$$

$$\alpha = \arctan\left(\frac{u_{ax}}{u_{tan} + \Omega r}\right) - \theta \tag{3}$$

The method was validated based on particle image velocimetry (PIV) measurements and CFD calculations of the MEXICO rotor by Herráez et al. (2018) for steady operation. They found a good performance of the method from $0.3\,R$ up to $0.9\,R$. In Rahimi et al. (2018), the model was further compared to alternative approaches applied to CFD simulations. Based on these comparisons and our specific focus on the dynamic change of inductions, rather than the total values, we considered the agreement at the root radius at $0.25\,R$ and the tip radius at $0.95\,R$ still reasonably good. Therefore we decided to include these radii in this analysis. However, they should be treated with care.

### 2.1.4 Ensemble averaging

Ensemble averages are used for the LDA data, hot wire and strain gauge measurements. The data of many repetitions is aligned, triggered by the pitch command. An average value at each time step is constructed out of this data, for the time span $-0.5\,\mathrm{s}$ to $3\,\mathrm{s}$, with the pitch step starting at $0\,\mathrm{s}$. For example, the ensemble average of the flapwise blade root bending moment $M_{\mathrm{flap}}$ is given by Eq. (4), with the counter of cycles $n$, total cycles $N$ and the time $t$.

$$M_{flap}(t) = \frac{1}{N} \sum_{n=1}^{N} M_{\mathrm{flap,\ single\ cycle}}^{(n)}(t) \tag{4}$$





This approach can smooth out non-deterministic variations and also structural interactions. In Fig. 6 a, the flapwise blade root bending moment for the step to low load is shown for single cycles and the ensemble average. As the pitch step is not aligned with the rotor position, the effect of the tower shadow, seen for the single cycles, is smoothed out for the ensemble average.

The high number of 900 repetitions, due to the nine different LDA positions with 100 pitch steps each, leads to a very small $95\%$ confidence interval (CI), which would barely be visible in the plots and, therefore, is not shown here or later load-related plots.

The induction factors have no fixed sampling frequency, as firstly, the underlying LDA measurements are non-equidistant and secondly, only values within the bisectrix of two blades are considered. To construct a single ensemble average out of this

data, we sort the 100 repetitions per LDA position to one signal and use a smoothing approach based on local regression and a weighted least squares and first order polynomial model. For the local regression, $1\%$ of the data (length of the total dataset is $4.5\,\mathrm{s}$) is used, whereas outliers get weight penalties and are not considered for more than six standard deviations. This filter is implemented as 'rlowess' within MatLab 2019b. This smoothed ensembled LDA based data is resampled to $1\,\mathrm{kHz}$, reducing the original non-equidistant data points by a factor of about 3. The sorted data points along the smoothed resampled signal and

$95\%$ CI for the axial rotor plane (rp) velocity at $0.7\,R$ for the step to low load are shown in Fig. 6 b.

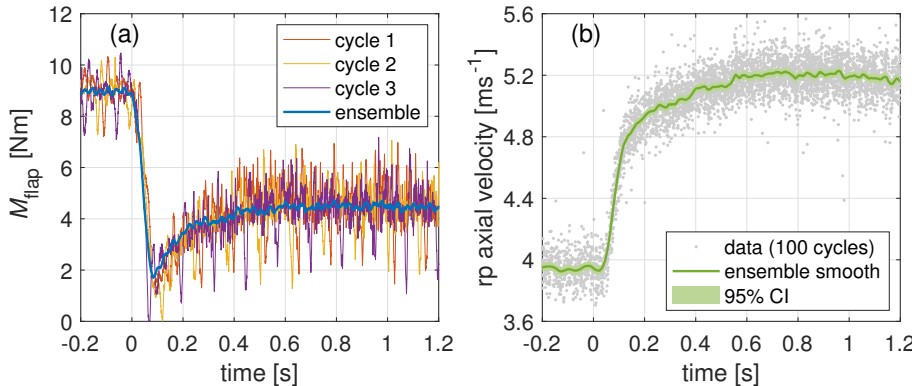

**Figure 6.** (a) Single cycles and ensemble average of flapwise blade root bending moment $M_{\mathrm{flap}}$. (b) Data points and smoothed ensemble average for axial rotor plane (rp) velocity in bisectrix of blades.

### 2.1.5  Corrections

We applied steady corrections to the thrust and torque signals. For the torque, the mechanical torque is measured at the torque meter. To obtain the aerodynamic rotor torque, we calibrated the friction in the bearings and the slip ring by running the drivetrain without blades with the motor, used as a generator in operation, and thus measured the friction with the torque meter.

We obtained a linear function of the angular rotor speed and added the respective value to the signal.

The rotor thrust is derived from the tower foot bending moment in fore-aft direction. The strain gauge was calibrated using defined forces in the thrust direction applied to the nacelle at the height of the rotor axis. The tower and nacelle drag was



experimentally calibrated with the turbine without blades and was subtracted from the signal. We used the free stream velocity ($u_0$) for this correction rather than a corrected wind velocity. This does lead to a small underprediction of the thrust. This error,

however is smaller as if we did not do a correction at all. In contrast to a dynamically corrected rotor plane wind velocity, this correction is a fixed value that does not influence the main shape of the dynamic load transient.

Dynamic corrections were considered for the torque and thrust signal. Directly after the pitch step, the torque control cannot keep the rotor speed completely steady, so there was a minor deviation of a maximum of $2\%$ of the rotor speed. We used Eq. (5) to correct the torque by the contribution $\Delta M$ associated to the angular acceleration $\dot{\Omega}$, where $I_{rot}$ is the equivalent rotational

inertia of the rotor and drivetrain.

$$\Delta M(t) = I_{rot} \cdot \dot{\Omega}(t) \tag{5}$$

After the pitch step, there is an oscillation of the tower, which is seen in the tower foot bending moment. We estimate the eigenfrequency of the tower and the damping constant of the oscillation iteratively and thus correct the measurement signal to obtain the aerodynamic thrust. The signals without the dynamic correction will also be shown in the results section as a

reference (see Fig. 20).

## 2.2 Time constant analysis

The decay process after the pitch step is investigated in terms of time constant analysis. Firstly, a one component time constant model (1c), like used by Schepers and Snel (1995) is applied, given by Eq. (6) for the arbitrary signal $S$.

$$S(t) = S_{t_0} - \Delta S \cdot \left(1 - \exp\left(\frac{(t_0 - t)}{\tau_{single}}\right)\right) \tag{6}$$

In Fig. 7 a, the fitting approach is outlined for an exponential transition to a higher value without an overshoot, representing the behaviour that is expected from an induction transient. Figure 7 b shows a signal with an overshoot and subsequent exponential decay to the new steady level. This represents the behaviour expected from a load. The fit starts when the pitch step is terminated at $t_0 = 0.070\,\mathrm{s}$ at the signal value $S_{t_0}$. The new steady level after the pitch step is $S_1$, being the mean value from $t = 2\,\mathrm{s}$ to $t = 3\,\mathrm{s}$. The difference $\Delta S$ is given by $S_{t_0} - S_1$ and also contains the information on the direction. The time constant

$\tau_{single}$ is fitted by means of the least root mean square error for the fitting range between $t_0 = 0.070\,\mathrm{s}$ and $t_{\mathrm{fit}} = 0.80\,\mathrm{s}$.

Secondly, we use a model with two time constants (2c), similar to Sørensen and Madsen (2006). The fitting model is given by Eq. (7). The fitting procedure is according to the single time constant model. However, three values are fitted, a fast time constant $\tau_{fast}$, a slow time constant $\tau_{slow}$ and the weighting factor $k$ of each exponential decay function, associated with the two time constants.

$$S(t) = S_{t_0} - \Delta S \cdot \left((1-k) \cdot \left(1 - \exp\frac{(t_0 - t)}{\tau_{fast}}\right) + k \cdot \left(1 - \exp\frac{(t_0 - t)}{\tau_{slow}}\right)\right) \tag{7}$$





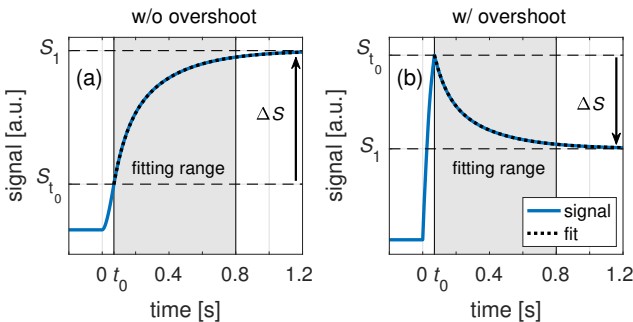

**Figure 7.** (a) Time constant fitting scheme for a signal with exponential decay behaviour without an overshoot. (b) Analogously for a signal with an overshoot.

### 2.3 Load reconstruction from induction measurement

Additionally to the strain gauge measured integral turbine loads, these loads are also reconstructed based on the induction measurements. The angle of attack along the rotor blade are already derived from the experiment through Eq. (3). The relative velocity $v_{rel}$ is given by Eq. (8). Hence, the information obtained by the momentum part of a BEM code is known from the

experiment. So we use the blade element theory (BET) part of a standard BEM code as outlined in detail by Hansen (2008). The force for the blade segments can be calculated for the normal direction according to Eq. (11) and for the tangential direction according to (12), where the lift force of the segment is given by Eq. (9) and the drag force by Eq. (10).

$$v_{rel} = \sqrt{u_{ax}^2 + (u_{ta} + \Omega r)^2} \tag{8}$$

$$F_L = \frac{1}{2} \cdot C_L(\alpha) \cdot \rho \cdot \Delta r \cdot c \cdot v_{rel}^2 \cdot F \tag{9}$$

$$F_D = \frac{1}{2} \cdot C_D(\alpha) \cdot \rho \cdot \Delta r \cdot c \cdot v_{rel}^2 \cdot F \tag{10}$$

$$F_N = F_L \cos\theta + F_D \sin\theta \tag{11}$$

$$F_T = F_L \sin\theta - F_D \cos\theta \tag{12}$$

$C_L(\alpha)$ and $C_D(\alpha)$ are the lift and drag coefficients for the respective angle of attack. These lift and drag polars are obtained by XFoil (see Drela (1989)). The blade segment width is $\Delta r$ and $c$ the chord length. The air density is given by $\rho$. The factor

$F$ accounts for the tip losses based on the Shen et al. (2005) tip loss model. The integral load signals are reconstructed by integration of the forces along the rotor blade.

The influence of unsteady airfoil aerodynamics (uA) on the blade level, namely the Theodorsen effect, is not contained in the axial wake induction and therefore has to be additionally considered in the reconstruction. We used the implementation given in detail in Pirrung et al. (2017). This is the inviscid part of the unsteady aerodynamics model by Hansen et al. (2004), which

treats the shed vorticity effects due to fast angle of attack changes as a time lag on the angle of attack $\alpha$. Thus, the magnitude and direction of the aerodynamic forces are influenced. The typical time lag of this uA effect is in the order of $c/v_{rel}$, whereas



the typical time constant of the dynamic inflow effect is $R/u_0$ and at least two magnitudes of size larger, as mentioned in Sect. 1. Reconstructed loads will be investigated with and without the uA model.

## 3 Results

Here the measurement results are described. In Sect. 3.1 the induction in the rotor plane is shown as a function of radial position determined along the procedure from Sect. 2.1.3. Then Sect. 3.2 shows the wake measurements from the hot wires as a function of streamwise position at hub height. Finally Sect. 3.3 presents the loads as measured from the straing gauges and the loads derived from the induction measurements according to the procedure of Sect. 2.3.

### 3.1 Induction results

In Fig. 8 a, the measured axial inductions, in Fig. 8 b, the tangential inductions and in Fig. 8 c, the derived angle of attack from the LDA measurements are presented for the steady high and low load cases. For the high load case, the axial induction

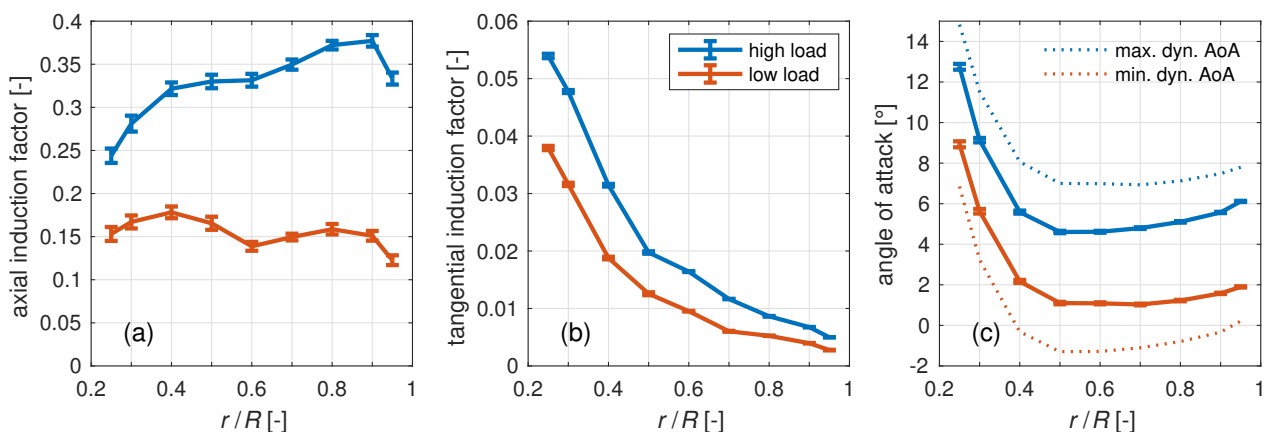

**Figure 8.** (a) Axial and (b) tangential wake induction factors and (c) derived angle of attack distribution from the LDA measurements for the steady high and low load case. The errorbars denote the $95\%$ confidence interval, based on the measured data for axial and tangential velocity in the rotor plane considering 100 measurement cycles.

has values between $0.25$ near the root ($0.25\,R$) and $0.38$ near the tip ($0.9\,R$), in general with an increasing trend with radius apart from the tip nearest radius. The low loaded case has a more uniform loading with values between $0.12$ and $0.18$. For comparison, the rotor equivalent axial induction obtained from momentum theory ($C_T = 4a(a-1)$) with $C_T$ based on the 245 strain gauge measurements do give similar values of $0.34$, respectively $0.14$.

The tangential induction for both cases is high near the root and decreases with radius with a high rate in the beginning and then more gentle. Due to the larger rotor torque, the high load case shows higher values.

The angle of attack distribution for the high load and low load case show angles of attack of $4°$ to $6°$ for the high load case from $0.4\,R$ on and $1°$ to $2°$ for the low load case. The values increase towards the root for both cases. We estimate the stall





angle as the angle of attack with the highest lift coefficient, where the flow is not completely separated. This angle for the used airfoils at the respective Reynolds number of the experiment is at $15°$ for the root airfoil used up to $0.4\,R$ and at $11.5°$ for the tip airfoil used from $0.5\,R$ on. Thus, the considered range of the blade is operating outside of the stall regime for both load levels for the steady states.

The difference between the two angle of attack distributions is smaller than the pitch step value of $5.9°$ the blades do
pitch, as the flow through the rotor and induction factors change between the two steady operational states. With these steady levels, the dynamic maximum and minimum angle of attack distributions can be estimated for an infinitely fast pitch step, only considering the influence of the wake. For this, we assume in a mind experiment that the flow field of the old steady state is unchanged, but the pitch step and thus geometrical change of the inflow angle is already done, giving us the extreme dynamic angles of attack. The flow field adapts to the new equilibrium and the new steady level just after the infinitely fast pitch step is
terminated.

These maximum and minimum dynamic angles of attack are shown in dotted lines in Fig. 8 c. For the step to high load the stall limit is approached at nearly $15°$ angle of attack at the root near radius of $0.25\,R$. For the step to low load, there is a minimum dynamic angle of attack of about $-1.5°$ in the middle of the rotor blade. This lowest angle of attack gives a lift coefficient of zero. For a finitely fast pitch step the flow already adapts during the pitch step and the extreme dynamic values
are closer to the new steady values. The uA effects further damp the overshoot of angles of attack. So apart from the blade root at the step to high load, where the stall limit is approached, the blade is operated outside the stall region for the pitch steps.

The axial induction factor transients are shown in Fig. 9 for four different radii ($0.3\,R$, $0.5\,R$, $0.7\,R$ and $0.9\,R$) for both pitch steps. They show direct evidence of dynamic inflow where the induction factors and therefore induced velocities reach the new equilibrium value only slowly.

The fits of the one (1c) and two time constant (2c) models are also shown in the plots. The fits start from the instance the pitch step is terminated at $t_0$. At that time the axial induction has already adapted by about $28\%$ on average of the difference between the steady axial induction levels for the radii from $0.3\,R$ to $0.9\,R$ independent of the pitch direction.

The fitted time constant $\tau_{single}$ of the 1c model is plotted over the radius for both pitch directions in Fig. 10. In the root near region up to $0.4\,R$ both pitch directions show similar values apart from the radius at $0.25\,R$, where the step to high load has a
higher time constant. For radii from $0.5\,R$ on the step to high load shows higher values than the step to low load. There is no clear trend obvious for the step to high load. In contrast, for the step to low load, there is a trend towards reduced time constants towards higher radii.

In Fig. 11, the three fitting parameters of the 2c model are presented. In the top row, the three fitted variables $k = k_{\mathrm{free}}$, $\tau_{fast}$ and $\tau_{slow}$ are plotted over the radius. Near the root at $0.25\,R$ the $k$ value for both pitch directions has a value of 1 respectively
nearly 1, indicating no contribution from $\tau_{fast}$. For radii up to $0.5\,R$ the values for the step to low load have a higher $k_{\mathrm{free}}$ value than for the step to high load, switching from $0.6\,R$ on to the tip. $\tau_{fast}$ does have similar values from $0.4\,R$ to the tip and for both pitch directions. Values towards the root are higher, however, the root nearest value is not relevant for the fit, as there $k_{\mathrm{free}}$ equals 1. Thus, the decay process is only defined by $\tau_{slow}$ there. For $\tau_{slow}$ there is no clear radial trend for both pitch directions, but a clear difference between pitch directions. For the step to high load $\tau_{slow}$ is higher for all radii than for the step

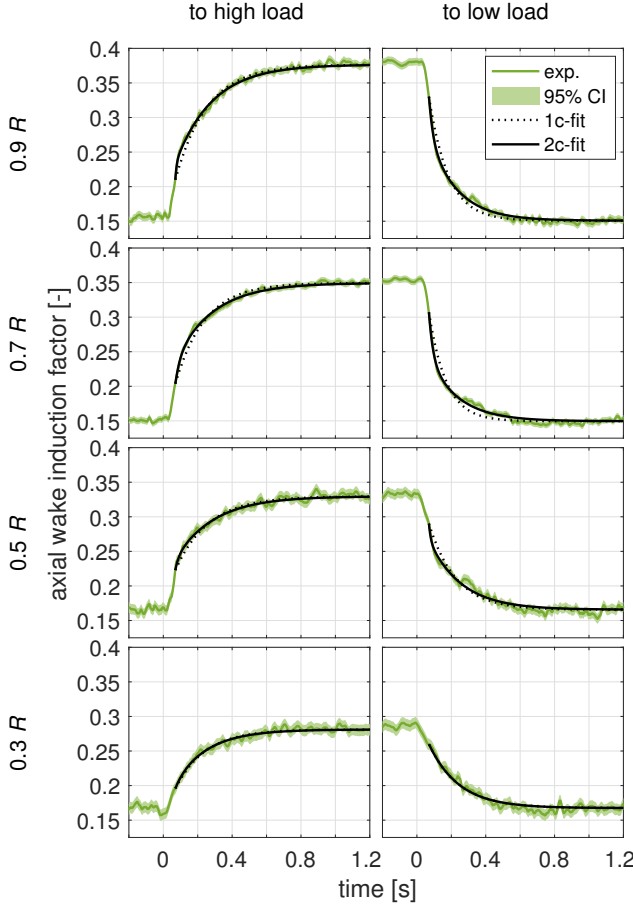

**Figure 9.** Axial wake induction factor over time for pitch step to high load and low load for the four radii $0.3\,R$, $0.5\,R$, $0.7\,R$ and $0.9\,R$ and the fitted exponential decay models with one time constant (1c) and two time constants (2c).

to low load. Due to the influnece of the varying weighting ratio of fast and slow time constant, a direct comparison of these time constants is limited.

To overcome this limitation, the ratio $k$ is fixed to a value of $k_{\text{fix}} = 0.79$, which is the mean value for all radii of both pitch directions of $k_{\text{free}}$. With this setting $\tau_{fast}$ contributes by $21\%$ to the decay of the axial induction. This fit is shown in the bottom row of Fig. 11.

The fitted $\tau_{fast}$ is high near the root for both pitch directions. For the step to high load $\tau_{fast}$ decreases from the root to $0.4\,R$, after which there is a slight increase again (ignoring an outlier at $0.8\,R$). For the step to low load $\tau_{fast}$ has nearly constant values from $0.5\,R$ to $0.9\,R$. Hence $\tau_{fast}$ is shorter for the negative load step, for radii larger than $0.5\,R$, which represents $75\%$ of the rotor swept area.

Values of $\tau_{slow}$ are slightly higher for the step to high load and show more variation than in the prior fit with $k = k_{\text{free}}$ ratio.
There is a slight radial trend to higher values. For the step to low load also more variation is apparent and a slight radial trend



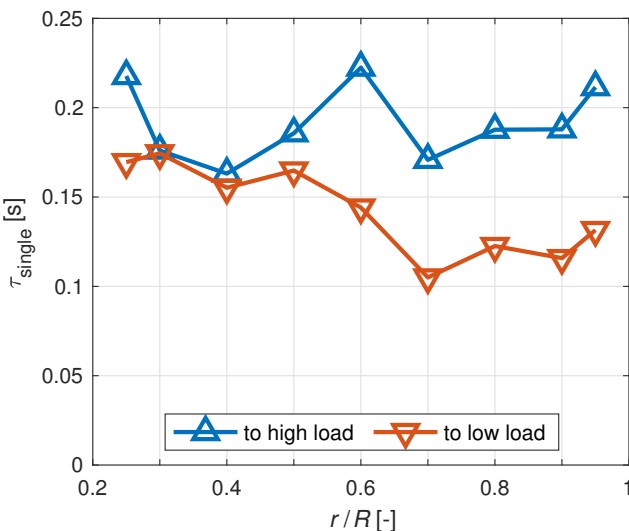

**Figure 10.** One time constant model fit of $\tau_{single}$ to the axial wake induction factor over the radius for both pitch directions.

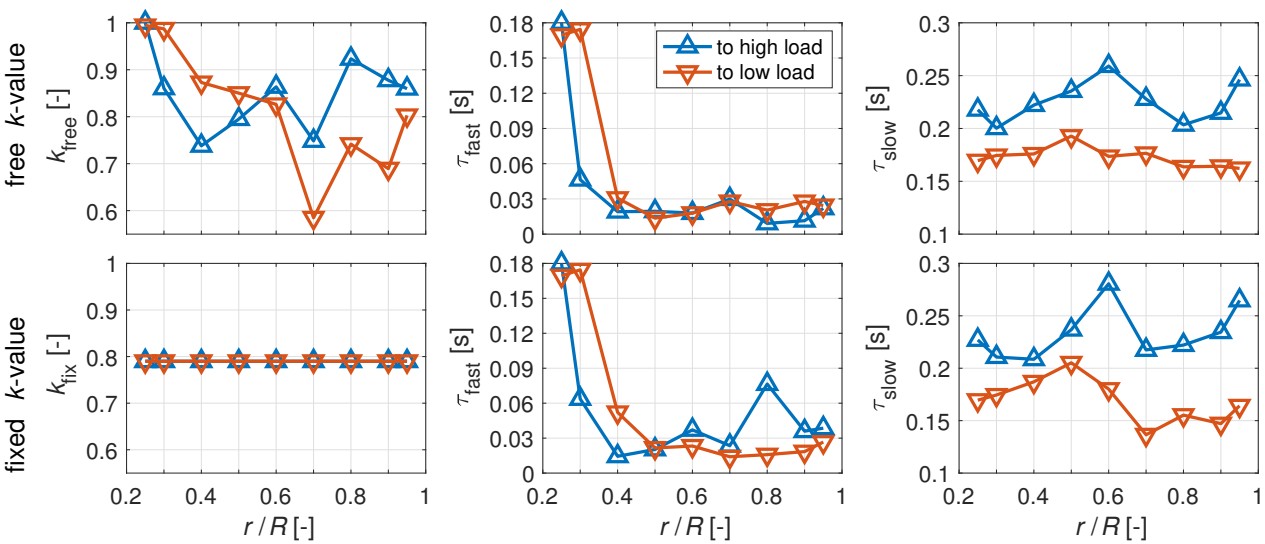

**Figure 11.** Two time constant model fit to the axial wake induction factor, derived from the rotor plane LDA measurements with the ratio $k$ and the fast $\tau_{fast}$ and slow $\tau_{slow}$ time constants. In the top row the k value is fitted as a free parameter. In the bottom row the weighting ratio of fast and slow time constant $k$ is fixed (to the mean value of both pitch directions and radii of the fit in the top row).





towards lower values is indicated. Taking the mean value over radius, the slow time constant for the step to low load is about $28\,\%$ lower.

The fitting accuracy of the applied models is determined based on the root mean square error (RMSE) in the fitting range $t_0$ to $t_{\text{fit}}$ between the measured signal and the respective fitted model and plotted over the radius in Fig. 12. For both step

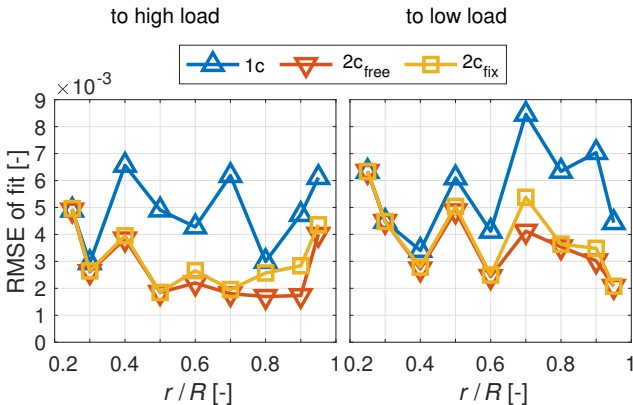

**Figure 12.** Root mean square error (RMSE) between measured axial wake induction factor and the three fitted models, 1c-fit and 2c-fit with $k_{\text{free}}$ and $k_{\text{fix}}$ for both pitch directions.

directions, there is no difference in RMSE for the root near stations up to $0.3\,R$ between the fitting models, which are the 1c and 2c model, once with $k = k_{\text{free}}$ and once with $k = k_{\text{fix}}$. For higher radii the error of the 1c fit is higher than for the two variants of the 2c model. The differences between the two variants of the 2c model are small, showing that there is only a small penalty for fixing the $k$ ratio .

The tangential wake induction factors over time for both pitch directions are presented for the four radii $0.3\,R$, $0.5\,R$,
$0.7\,R$ and $0.9\,R$ with the fit of the 1c model in Fig. 13. In contrast to the axial induction, the tangential induction shows an overshooting behaviour. The exponential fit starts around $t_0$ at the respective minimum or maximum peak, as this showed to improve the fitting. The starting point of the fit thus varies between $0.059\,\text{s}$ and $0.095\,\text{s}$, with a mean value of $0.074\,\text{s}$. The overshoot, in general, seems more prominent for the step to low load and only barely present within the signal noise at the radius of $0.7\,R$ for the step to high load.

The fitted $\tau_{single}$ values to the 1c model are plotted over radius for both pitch directions in Fig. 14. Cases where the overshoot is smaller than three standard deviations of the filtered signal of the new equilibrium were excluded due to a very high sensitivity on the starting point of the fit. For the step to high load, only two radii at the root and two radii near the tip fulfil this requirement, whereas for the step to low load only the tip most radius at $0.95\,R$ is excluded. For the radii where values for both pitch directions are available, the step to high load shows higher $\tau_{single}$ values. For both pitch directions, $\tau_{single}$ is lower than
the corresponding $\tau_{single}$ for the axial inductions. We do see no connection of this unexpected overshooting behaviour of the tangential induction to the slight rotor speed deviations, which are present between $0\,\text{s}$ and $0.6\,\text{s}$ and thus at a time frame up to one magnitude of order higher.

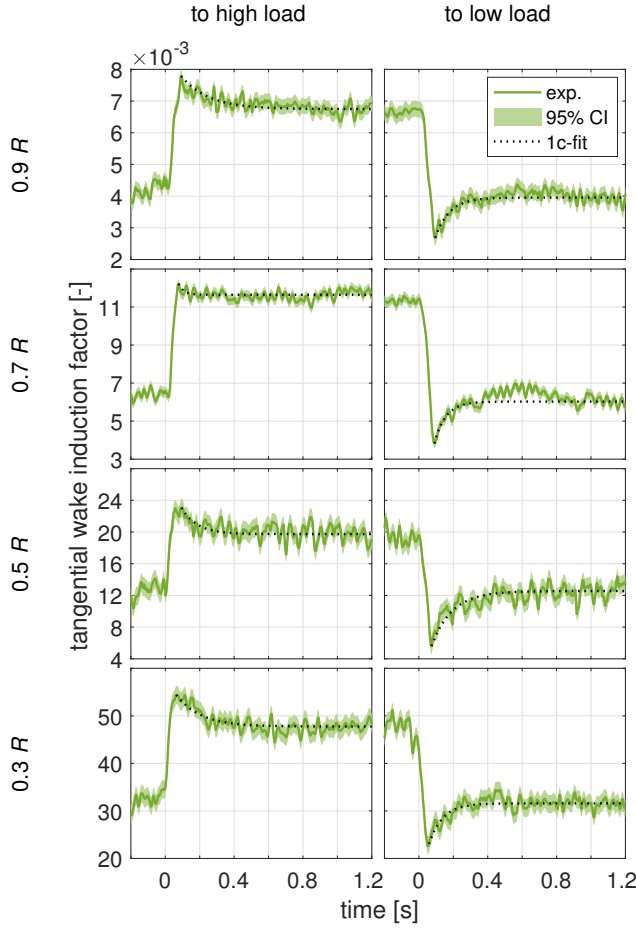

**Figure 13.** Tangential wake induction factor over time for pitch step to high load and low load for the four radii $0.3\,R$, $0.5\,R$, $0.7\,R$ and $0.9\,R$ and the fitted exponential decay model with 1 time constant (1c).

## 3.2 Wake results

In the following the hub height hotwire measurements downstream of the turbine in the near and close far wake up to $2D$
downstream are analysed. The wake measurements are shown as a surface plot for four different timestamps in Fig. 15, for both pitch directions and normalised by the free stream velocity. The first contours at $t = 0\,\mathrm{s}$ are the starting equilibrium condition. For the step to high load, the initial wake, being the steady low load case, shows wake velocities around $0.8\,u_0$ coloured in green tones for $0.2\,R$ to $1\,R$ radial positions. For the step to low load, the initial velocity field in the wake, being the steady high load case, is around $0.5\,u_0$, cloured in blue tones. The dotted red line is the isoline of $0.9\,u_0$, which we use as
an indication of the wake streamtube.

The contours for the following time stamps show the transition to the new steady state. These transitions seem at first glance different for the two pitch directions. For both pitch directions at $t = 0.4\,\mathrm{s}$ and at around $1\,D$ to $1.25\,D$ there seems to be the



eawe
european academy of wind energy

WIND
ENERGY
SCIENCE
DISCUSSIONS

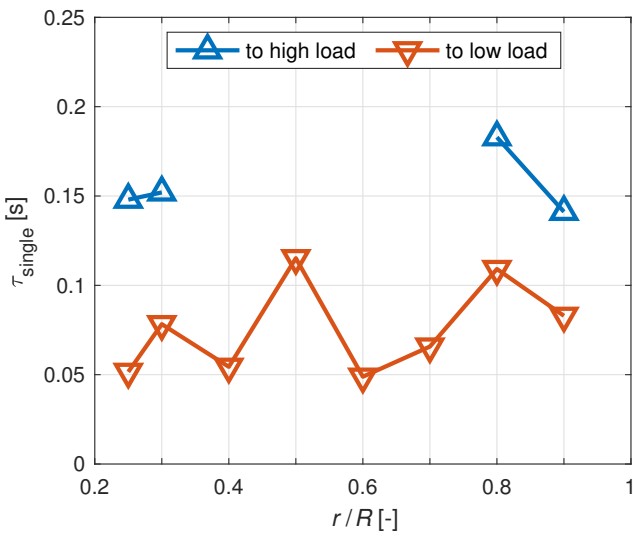

**Figure 14.** One time constant model fit of $\tau_{single}$ to the tangential wake induction over the radius for both pitch directions.

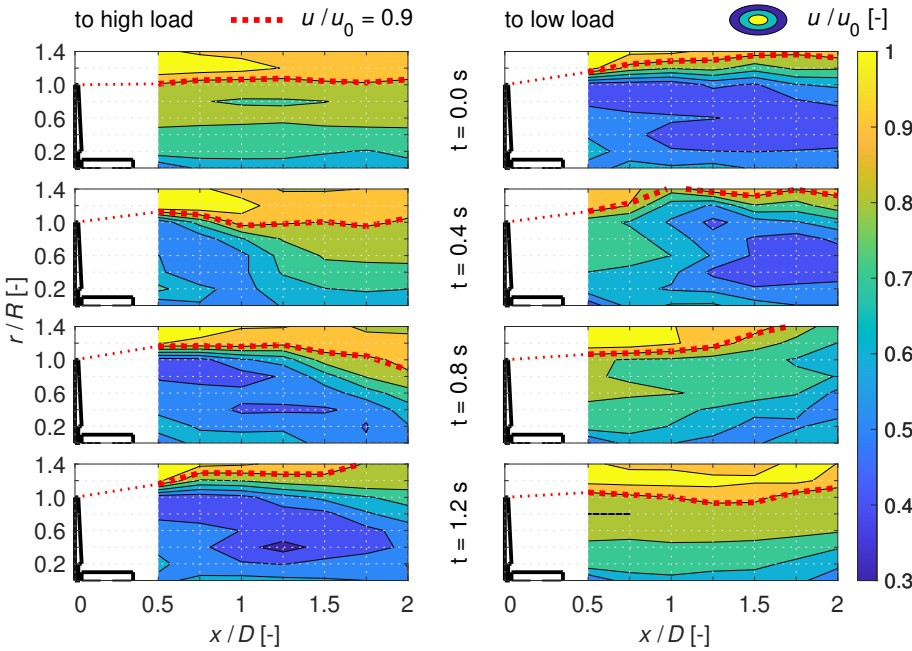

**Figure 15.** Velocity contour in the horizontal plane at hub of the wake at the four different time stamps, $0\,\text{s}$, $0.4\,\text{s}$, $0.8\,\text{s}$ and $1.2\,\text{s}$ after the pitch steps to high and to low load, normalised by the free stream velocity. The turbine dimensions are indicated in correct scale in the plots as a reference, with the x-axis being the axis of rotation.





transition line where old and new wake meet. The indicated wake radius shows, that the dynamic wake streamtube is constricted at $1\,D$ for the step to high load, whereas it is widened up for the step to low load. This behaviour is opposite to what we do

expect for the new steady streamtubes, where for the step to high load, the wake widens, due to the higher thrust. Further, it narrows due to the lower thrust for the step to low load. For the time stamp at $t = 0.8\,\mathrm{s}$ a similar picture can be seen where this transition between old and new wake has progressed to around $1.75\,D$. The figure at the left bottom is very similar to the figure at the right top and the same holds for the figure at the right bottom and left top. This indicates that the new equilibrium is approached at $t = 1.2\,\mathrm{s}$.

To further interpret this measurement, normalised velocity contours are presented as the difference to the new equilibrium steady state in Fig. 16. Therefore, a value of $0.5$ means that the wake has to adapt by $0.5\,u_0$ to reach the new equilibrium.

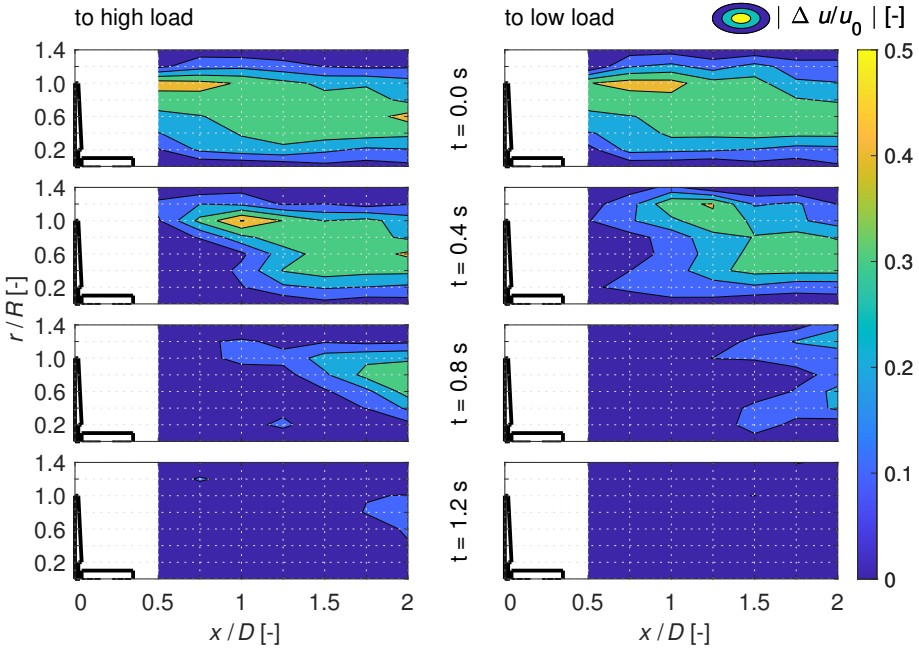

**Figure 16.** Velocity contour in the horizontal plane at hub of the difference to the new steady wake equilibrium at the four different time stamps, $0\,\mathrm{s}$, $0.4\,\mathrm{s}$, $0.8\,\mathrm{s}$ and $1.2\,\mathrm{s}$ after the pitch steps to high and to low load, normalised by the free stream velocity. The turbine dimensions are indicated in correct scale in the plots as a reference, with the x-axis being the axis of rotation.

The starting conditions for both pitch directions look alike, as they show the difference between the two steady states. At the timestamp of $t = 0.4\,\mathrm{s}$ we see a very similar shape of this potential like area for both pitch directions. A widening of the wake for the step to low load is indicated when concentrating on the orange field near the tip radius, which is at a radial position of

$1\,R$ for the step to high load and at $1.2\,R$ for the step to low load. For the step to low load, this orange maximum has travelled further than for the step to high load. At the timestamp at $t = 0.8\,\mathrm{s}$ the wake for the step to low load has adopted more to the new steady state than for the step to high load. At the timestamp of $t = 1.2\,\mathrm{s}$ for the step to low load, the equilibrium is reached. For the step to high load, the wake has not completely adapted to the new equilibrium with still a clear lighter blue area at $2D$.



Next, we analysed the deployment of the axial wake velocity as a response on the load steps. Six of the hot wire signals used to make the contour plot are plotted over time for the step to high load for the radii of $0.2\,R$, $0.6\,R$ and $1\,R$ at $0.5\,D$ and $1.5\,D$ distance behind the turbine are presented in Fig. 17. The 1c model fit is applied to the hot wire signal. The fit does start at $t_0$,

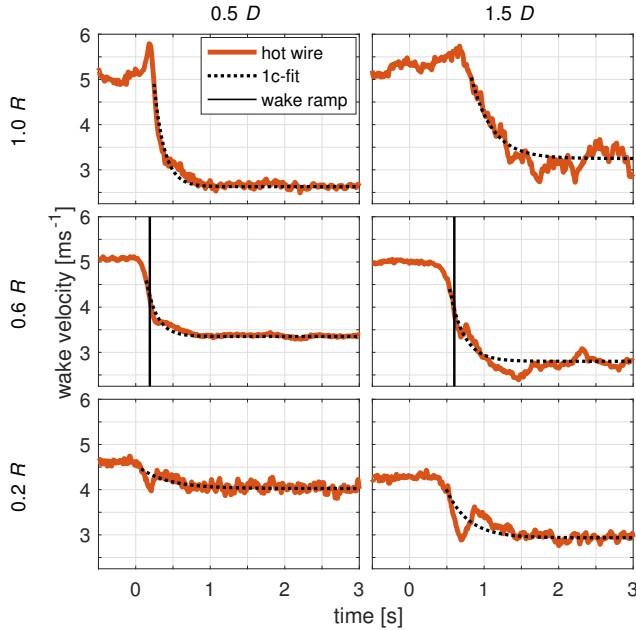

**Figure 17.** Ensemble averages of the axial wake velocity measured by hot wires for the three radii of $0.2\,R$, $0.6\,R$ and $1\,R$ at $0.5\,D$ and $1.5\,D$ distances behind the turbine for the step to high load.

which is defined at the point where the velocity has adjusted by $28\,\%$ to the new equilibrium. This definition is based on the mean value the axial inductions have adjusted during the pitch step. For the radius at 0.6 R a vertical line, called wake ramp, is drawn at $t_1$, where the signal has adapted by $50\,\%$. For the radius of $0.2\,R$ the velocity transient shows a local structure, where the velocity quickly decreases, increases again to subsequently decrease to the new equilibrium for both shown distances. At the radius of $0.6\,R$ the signal decays exponentially. At 1 R the signal increases quickly to a peak at nearly free stream velocity before decreasing exponentially to the new steady value. The signals at $0.5\,D$ behind the turbine for the two higher radii, $0.6\,R$ and $1\,R$, show a better fit with the exponential decay function. For the farther distance at $1.5\,D$ the hot wire signals show an overshoot to lower velocities than the new equilibrium at around $1.5\,\mathrm{s}$ to $1.7\,\mathrm{s}$ for the two larger radii.

Further, we compared the fitted $\tau_{single}$ values for the wake flow measurements for both pitch directions. They are presented in Fig. 18. Between $0.5\,D$ and $1.5\,D$ no clear differences can be seen between pitch directions, apart from the root nearest radius of $0.2\,R$. The fitting of this radius is very sensitive due to the described local structure within the signal and should not be overinterpreted. For farther distances behind the turbine at $1.75\,D$ and $2\,D$ there is a clear difference between the pitch directions, with noticeable higher time constants for the step to high load for the radial range of $0.4\,R$ to $1\,R$, compared to both



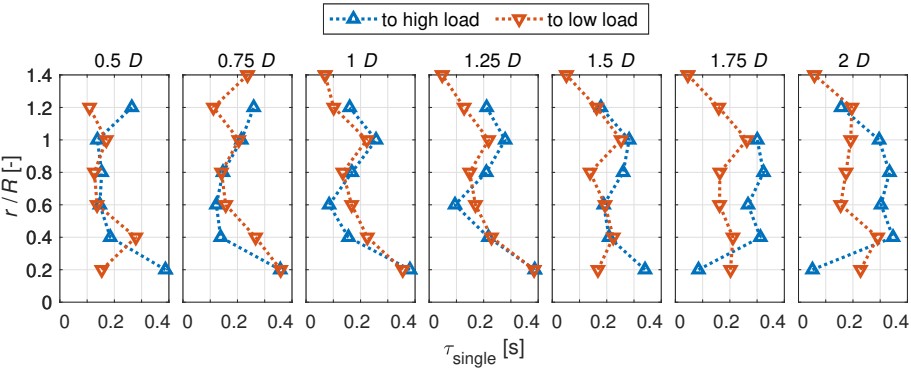

**Figure 18.** Spanwise evolvement of the axial wake velocity from $0.5\,D$ to $2\,D$ parametrized by the fitted time constant of the one time constant model for both pitch directions

the time constants for the step in the same direction at rotor nearer distances and also in comparison to the step to low load at the same distances.

In the next step, we analyse the wake ramp velocity to measure how fast the transition point (the wake ramp) between the old wake and the new wake does convect. We define it as the velocity this wake ramp travels from one considered downstream position to the next. So in Fig. 17 at $0.6\,R$ we do have the wake ramp for $0.5\,D$ at $t_{\mathrm{ramp},0.5\mathrm{D}} = 0.20\,\mathrm{s}$ and for $1.5\,D$ at $t_{\mathrm{ramp},1.5\mathrm{D}} = 0.56\,\mathrm{s}$. Within this time difference, the wake ramp thus has travelled by $1\,D$, giving a mean wake ramp velocity between these downstream distances.

In Fig. 19, this wake ramp velocity is shown for both pitch directions, normalised by the free stream velocity. The radii of

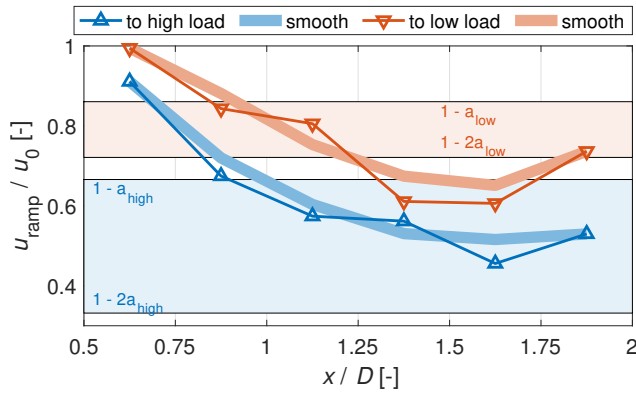

**Figure 19.** Velocity of the wake ramp, defined by the mid point of the swept area weighted mean of the radii $0.4\,R$, $0.6\,R$ and $0.8\,R$, for both pitch directions, normalised by the free wind velocity. For orientation also the theoretical normalised wind velocity in the rotor plane (1-a) and in far wake (1-2a), based on the measured thrust coefficient, are indicated for both steady states.

$0.4\,R$, $0.6\,R$ and $0.8\,R$ were considered and a swept area-weighted mean, considering an annulus of $\pm\,0.1\,R$ for each radius,



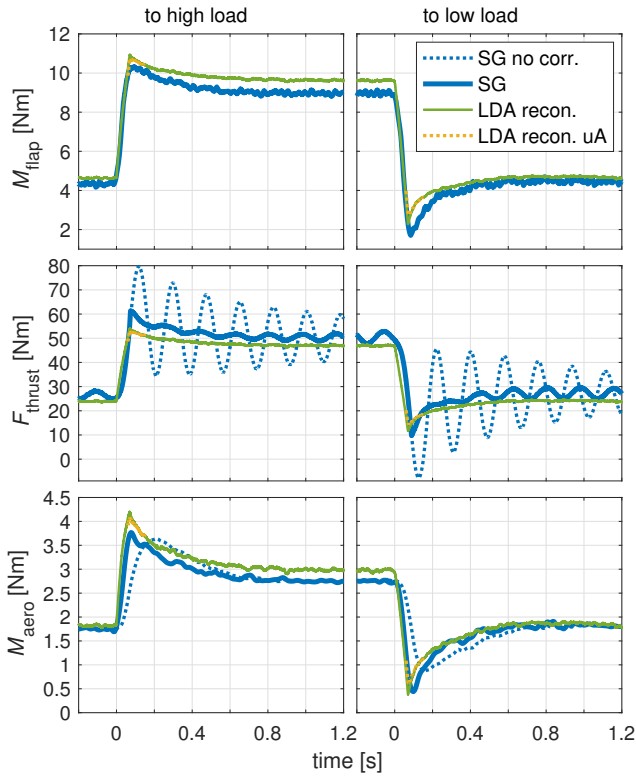

**Figure 20.** Integral turbine loads based on strain gauges with (SG) and without (SG no corr.) the dynamic corrections (as introduced in Sect. 2.1.5) and reconstructed loads based on Blade Element Theory from the LDA measurements without and with uA model.

was used. Thus between every two considered downstream distances a mean velocity with which the wake ramp moved can
be calculated. Between the two nearest distances to the rotor, $0.5\,D$ and $0.75\,D$, the wake ramp convects for both cases with
$0.9\,u_0$ to $u_0$. This is faster than the expected velocity in the rotor plane for even the low loaded rotor. With increasing distance
to the rotor for both cases the wake ramp velocity decreases up to $1.5\,D$, whereas the velocity is higher for the step to low load.
Based on this defined wake ramp velocity the wake convection is on average $26\%$ faster for the step to low load.

### 3.3   Loads results

Next, we compare the integral loads shown in Fig. 20. Two independent methods have been used. Once measured directly
with strain gauges (SG) and once obtained indirectly from the LDA measurements in the rotor plane with the reconstruction
procedure given in Sect 2.3 (LDA recon.). For the latter we distinguish between the use of the uA model and without it. For
the strain gauge measurements additionally a version without the dynamic corrections for $F_{\mathrm{thrust}}$ and $M_{\mathrm{aero}}$ (SG no corr.) as
introduced in Sect. 2.1.5 is presented to show the raw data. For the reconstructed loads further the influence of unsteady airfoil
aerodynamics (uA) effects is shown around the overshoot, where it differs from the case without the uA model.



Firstly, a clear overshooting behaviour of all load signals is apparent. Comparing the strain gauge measurements for $M_{\text{flap}}$, $F_{\text{thrust}}$ and $M_{\text{aero}}$ with the signals reconstructed from the LDA measurements indicate a good match of signals by means of steady values as well as the dynamic overshoot. Considering the uA model leads to a peak shaving of the overshoot. Also, the overshoot peak for the strain gauge signals, as well as for the reconstructed value with the uA model, are slightly shifted to

higher $t_0$ values.

A more detailed comparison of the steady values at high and low load between the strain gauge measured integral loads and the reconstructed loads is plotted in Fig. 21, alongside the deviations of the reconstructed loads to the strain gauge measured loads. The two methods give similar load levels. Deviations differ between the load signals. The good agreement in terms of steady loads indicates good performance of this reconstruction approach with maximum deviations of 11%. We attribute the

slight overprediction of reconstructed loads for the $M_{\text{flap}}$ and $M_{\text{aero}}$ on the one hand and the underprediction of $F_{\text{thrust}}$ on a higher weighting of the higher radii for the first two signals than for the $F_{\text{thrust}}$ signal.

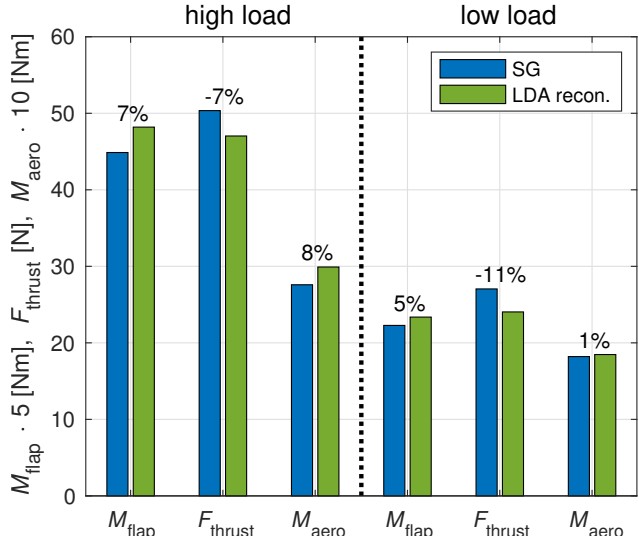

**Figure 21.** Comparison of the steady load levels for high and low load for the integral turbine loads $M_{\text{flap}}$, $F_{\text{thrust}}$ and $M_{\text{aero}}$ obtained by strain gauges and reconstructed from LDA measurements.

Further, the dynamics after the pitch step are compared. The amount of load overshoot, normalised by the difference between the steady values, is given for the load signals for both pitch directions in Fig. 22 a. The addition of the uA model reduces the amount of overshoot for all load channels. For $M_{\text{flap}}$ the reconstructed loads provide a good fit, whereas the values with uA

model show differences. High deviations are seen for $F_{\text{thrust}}$ between strain gauge measurement and reconstructed loads, especially with the uA model, but also without the model, for both pitch directions. The torque $M_{\text{aero}}$ shows a good match with the reconstructed loads for both pitch directions.

The amount of overshoot is higher for the step to low load, when comparing the amount of overshoot between the two pitch directions per load channel and method. Based on the LDA reconstructed loads, which are not subject to dynamic corrections





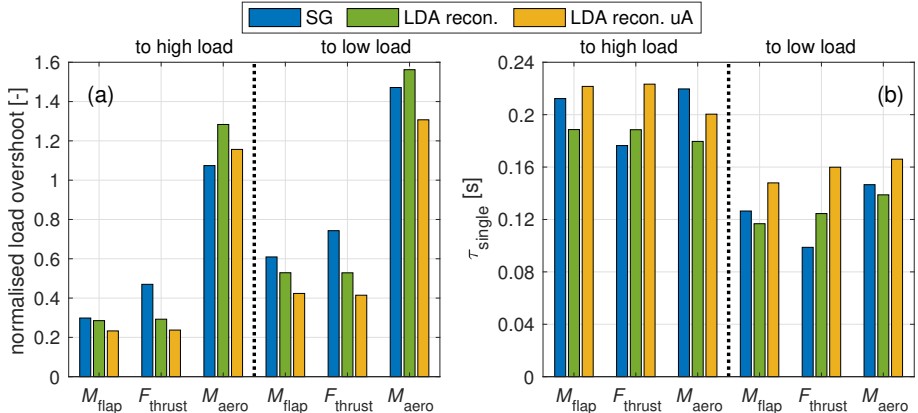

**Figure 22.** (a) Comparison of the amount of overshoot of the integral turbine loads $M_{\text{flap}}$, $F_{\text{thrust}}$ and $M_{\text{aero}}$, obtained by strain gauges and reconstructed from LDA measurements without and with uA model for both pitch directions. (b) Comparison of the $\tau_{single}$ fit of the one time constant model to these integral loads.

that might introduce errors, the normalised overshoot for the $M_{\text{flap}}$ and $F_{\text{thrust}}$ is similar per pitch direction, whereas the overshoot in $M_{\text{aero}}$ is 3 to 4.5 times higher.

The fitted values for the one time constant model to the integral loads is presented in Fig. 22 b for both pitch directions. The time constants for all load signals and methods are longer for the step to high load than to low load. This is consistent to what was found in the velocity results before. The time constant is in general increased by including the uA model. For the step to high load $M_{\text{flap}}$ and $M_{\text{aero}}$ show a good match with the reconstructed uA method and an acceptable match for the step to low load. For $F_{\text{thrust}}$ there are higher deviations between the strain gauge based time constant and the time constant based on the reconstructed uA load.

The theoretical maximum overshoot based on the steady induction measurements and the theoretical dynamic maximum and minimum angle of attack distribution, as was presented in Fig. 8, is investigated. The steady integral loads for high and low load are reconstructed as before for Fig. 20. The maximum load overshoots are obtained by using the maximum and minimum dynamic angle of attack distribution with the inductions for the steady state before the pitch step, so an infinitely fast pitch step is assumed and no uA effects that showed to reduce the overshoot, are considered. The derived theoretical maximum amount of overshoot is plotted in Fig. 23. A clear difference can be seen between the pitch directions for the axial loads $M_{\text{flap}}$ and $F_{\text{thrust}}$, with about twice as high an amount of overshoot for the step to low load, than for the step to high load. For the torque the general trend is the same with 2.5 to 4.5 times the overshoot in comparison to the other signals. The trend is similar to the repeated LDA recon. values in the plot.



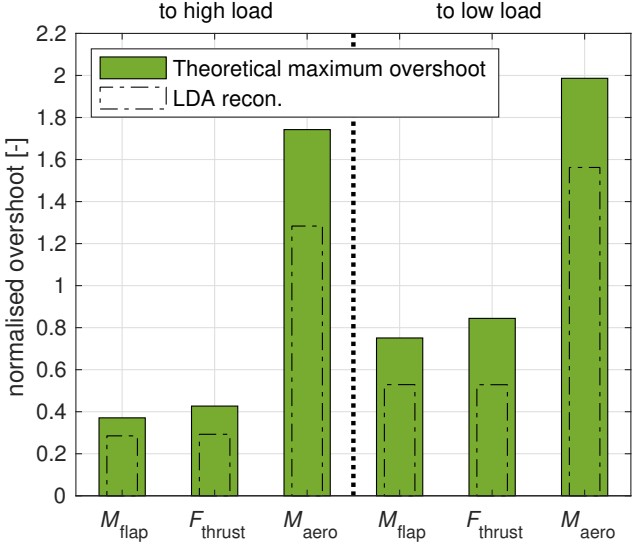

**Figure 23.** Theoretical dynamic overshoot for the step to high and low load for the integral turbine loads $M_{\text{flap}}$, $F_{\text{thrust}}$ and $M_{\text{aero}}$, based on reconstruction from the LDA data, assuming an infinitely fast pitch step in comparison to the LDA recon. case from Fig. 22 a.

## 4  Discussion

In this section, the results for the inductions, wake flow and loads will be discussed. A focus for comparisons is on publications in connection to the NREL unsteady aerodynamics experiments phase VI (see Hand et al. (2001)), later referred to as just phase VI, as this experiment is the most widely studied dynamic inflow dataset so far.

### 4.1  Inductions transients

Dynamic inflow phenomena on wind turbines were already indirectly shown in experiments based on integral loads for the 2 MW Tjæreborg turbine (see Øye (1991)), based on pressure sensor derived sectional forces on the phase VI (see Hand et al. (2001)) and MEXICO (see Boorsma et al. (2018)) model turbines and in the wake measurements behind an actuator disk with dynamic thrust changes (see Yu et al. (2017)). The presented axial induction transients in Fig. 9 based on the LDA measurements in the rotor plane are the first direct experimental evidence of dynamic inflow phenomena for wind turbines.

Pitch steps of the phase VI experiment were performed at $5\,\text{ms}^{-1}$ wind speed and a tip speed ratio (TSR) of 7.5. The pitch step of $15.9°$ took place during approximately 1/3 rotor revolution (pitch rate of $57°/\text{s}$ over 0.28 s). This extreme pitch step is between a very highly loaded rotor at a rotor equivalent axial induction of 0.5 and 0 for the unloaded rotor. At the highly loaded rotor state, the turbine is in the turbulent wake state. This high induction state was later suspected to be responsible for problems in validation (Sørensen and Madsen (2006)).





Pitch steps in this paper were performed at $6.1\,\mathrm{ms}^{-1}$ and a TSR of 7.4. The pitch step of $5.9°$ was performed during approximately 1/2 rotor revolution between rotor equivalent inductions of $0.34$ and $0.14$. The experiment thus is not operated in the turbulent wake state at the high load case.

The observed slight radial dependency for the single time constant (see Fig. 10), where the values slightly decrease towards the tip for the step to low load is in accordance with the observation by Schepers (2007) for the phase VI experiment, despite the more extreme change in rotor loading.

Pirrung and Madsen (2018) reproduced this behaviour for the phase VI experiment in a cylindrical wake model simulation. They explain this observation by the nature of the dynamic inflow process that cannot be described correctly by a one time

constant model but by a two time constant model. They reason that the axial induction adapts by different amounts during the pitch step depending on pitch direction. This is governed by the wake velocity before the pitch step and thus influences the one time constant fit to the normal forces after the overshoot. In contrast, in the presented axial induction transients (see Fig. 9), we see that it adapts similarly during the pitch step for both pitch directions, by $28\%$ on average.

For the two time constant model fit to the axial wake inductions two variants, once with a freely fitted weighting ratio

$k_{free}$ and once with a prescribed weighting ratio $k_{fix}$ of time constants, were shown in Fig. 11. Using $k_{fix}$ allows for a direct comparison of $\tau_{fast}$ and $\tau_{slow}$ between the pitch directions. The radial dependency of $\tau_{fast}$, with high values towards the root extends further into the blade for the step to low load. In their investigation of the normal force measurements on the phase VI experiment, Sørensen and Madsen (2006) also saw this high $\tau_{fast}$ values at the root nearest radius of $0.3\,R$ for the step to low load, but not for the step to high load. In our and the phase VI experiment, this high $\tau_{fast}$ near the root is of the order of $\tau_{slow}$.

In our experiment, there is a slight difference in the $\tau_{fast}$ between pitch directions, with lower values for the step to low load from $0.5\,R$ to the tip. This difference between the pitch directions is not seen by Sørensen and Madsen (2006).

For $\tau_{slow}$, a very slight radial dependency can be seen for the fit with $k_{fix}$, with slightly increasing values towards the tip for the step to high load and slightly decreasing values for the step to low load, similar to the fit of $\tau_{single}$.

The $\tau_{slow}$ values are on average about $28\%$ faster for the step to low load, than for the step to high load. The same trend

can be seen in Sørensen and Madsen (2006), however by a much higher value of around $100\%$. The likely explanation is the higher difference in axial induction between the steady load levels for the phase VI experiment. Sørensen and Madsen (2006) and Pirrung and Madsen (2018) discuss this scaling of the time constants with the wake deficit and thus mean axial induction of the rotor. Also, the Øye dynamic inflow model (see Schepers and Snel (1995)) and other recent dynamic inflow models (e.g. Madsen et al. (2020)) use the axial induction to scale the time constants. There the fast time constant represents the near wake

dynamics and decreases with radius and the slow time constant represents the far wake dynamics

Sørensen and Madsen (2006) and Pirrung and Madsen (2018) elaborated, based on measurements and simulations, that a two time constant model better describes the dynamic inflow process than a one time constant model. The two time constant approach also was implemented in the Øye dynamic inflow model (see Schepers and Snel (1995)). Yu et al. (2019) further showed, based on actuator disk vortex models, that a two time constant model describes the process better than a one or

three time constant model. Comparing the fitting error of the 1c and 2c models to the axial wake induction measurements of





our experiment (see Fig. 12), we can verify this finding based on the first direct meaurements of dynamic inflow phenomena. However, we found the root region to be an exception, where the process is defined by only one time constant.

The only minor differences in fitting error between the two variants with $k_{free}$ and $k_{fix}$ (see Fig. 11), illustrate the high sensitivity of the fitting process. Small changes in $k$ have a noticeable impact on the $\tau_{fast}$ and $\tau_{slow}$ values, where one increases

as the other decreases.

The overshooting behaviour of the tangential induction is a new finding. The time constants of the one time constant model have no clear radial dependency. They are lower for the step to low load and in general, slightly lower than those fitted for the axial induction. The overshoot also is more prominent for the step to low load. This behaviour is of interest for the physical understanding of the dynamic inflow effect, as the overshoot in the torque is directly counteracted by the change in

wake rotation. We assume that the shed vortices due to the change in circulation introduce this instant overshoot in tangential induction. For the modelling of the dynamic inflow effect in BEM based codes, this behaviour is only of secondary interest, as the influence of the tangential induction on the angle of attack is negligible, apart from the blade root.

## 4.2    Wake evolution

We did not see a relevant influence of the shear layer between the open jet wind tunnel and the surrounding air in the wake

snapshots (see Fig. 15). On that basis, we decided to include these measurements in the paper. The main near wake and beginning far wake dynamics due to the pitch step should not be disturbed, despite the high blockage ratio of $0.28$, as long as the wake and the shear layer of the open jet wind tunnel flow do not interact.

The velocity snapshots in the wake show differences in wake evolution between the pitch directions. The faster progression of the wake for the pitch step to low load, supports the presumption made by Schepers (2007) of different convection velocities

of the wake consisting of the old vorticity and new vorticity, depending on the pitch direction. We assume the faster convection of this mixed wake for the case of the step to low load, starting with a low wake velocity pushed by a higher wake velocity to be the main driver of the faster time constants ($\tau_{single}$ and $\tau_{slow}$) of the axial wake induction.

The dynamic widening of the wake after the pitch step to low load, as drawn in Fig. 15, is a new finding. We attribute it to the sudden change in trailed vorticity shed from the blade tip region. Part of the slow old wake is then pushed outboard by the

fast new wake. The wake is not accelerated by the fast new wake at these larger radii, but by the even faster free stream wind velocity. This behaviour is a possible explanation for the decreasing time constants $\tau_{single}$, respectively $\tau_{slow}$, towards the tip for the pitch step to low load.

The overshoot in the velocity in the wake after a sudden change in thrust, as observed here for the radii at $1.0$ R for the step to high load (see Fig. 17), has been discussed by Yu et al. (2017) based on wind tunnel experiments with a variable porosity

actuator disk. They attribute these overshoots to the shed vorticity at the actuator disk edge due to the fast thrust change. The local structure at $0.2\,R$ where the root transition towards the axis of rotation starts, is opposite to the overshoot at $1\,R$ for the step to high load. We assume this to be the counterpart, the shed vorticity at the blade root, due to the sudden change in trailed vorticity. A connection of this shed root vortex to the radial dependency of $\tau_{fast}$ near the root cannot be concluded from the data but seems possible.





In the time constant analysis of the hot wires a clear difference between pitch directions is only apparent for the two farthest distances from the rotor at $1.75\,D$ and $2\,D$, where the step to low load leads to faster time constants. In the actuator disk experiments by Yu et al. (2017) this trend also seems to be more prominent for the higher distances from the actuator disk.

    The quantitative comparison of the wake ramp velocity (see Fig. 19), as a measure of the convection of the transition point between old and new wake, shows the faster velocity for the step to low load. On average, this wake convection velocity is

$26\%$ faster for the step to low load than for the step to high load. This difference resembles the difference in $\tau_{single}$ and $\tau_{slow}$ from the induction investigation in Sect. 4.1.

    We see a fast initial wake ramp velocity between $0.5\,D$ and $0.75\,D$, which is near the free stream velocity and higher for both cases than the expected axial wind velocity in the rotor plane for the low loaded rotor. An obvious explanation is that in the near wake, the dynamic inflow process of the whole rotor is governed by the influence of the trailed tip vortex, which does

convect at similar velocities.

    In the further course, the wake ramp velocity slows down for both cases. For the step to low load, it even slows down between $1.25\,D$ and $1.75\,D$ to a lower value than the far wake velocity according to momentum theory for the new steady state. This behaviour can be explained by the mixed wake of both the old and new wake, which influence each other.

### 4.3   Load transients

The general comparison of the load signals obtained from strain gauges and reconstructed from the LDA measurements shows a good agreement and thus proved the physical consistency of the induction measurements for these dynamic experimental cases.

    For the steady equilibrium states, the quantitative comparison between the strain gauge measurements and the reconstructed loads yields in generally good agreement, apart from the thrust at low load, which we suspect to be connected with the airfoil

polars, especially at the root airfoil.

    For the flapwise blade root bending moment and the thrust, the load overshoot is much more pronounced for the step to low load than for the step to high load, whereas the trend for the torque is the same, but to a smaller extent. The magnitude of the overshoot depends on how much the axial induction has already adapted during the pitch process. As discussed in Sect. 4.1 they adopt a similar amount during the pitch step for both pitch directions, so that this cannot explain the difference in overshoot

between the pitch directions. The more pronounced overshoot of loads is also discussed for the phase VI measurements in Schepers (2007). He suspected the high angles in the stall region at the highly loaded rotor blade to be a possible reason for that phenomenon. In contrast, the high load case is operating safe outside of the stall regime for the experiment presented here, apart from the very root.

    The simplified approach to estimate the theoretical maximum overshoot for the load signals based on the steady inductions

and the pitch angle change with the BET reconstruction (see Fig. 23), provides a solid reason for the different amount of load overshoot. It only depends on the turbine load characteristic, as the theoretical overshoot show a very similar ratio between load channels and pitch directions to the actual measured and reconstruceted load overshoot. The angle of attack overshoot for





both pitch directions is the same in this mind experiment, however, near the high load the turbine reacts less sensitive on angle of attack changes than at the low load state.

Further, in the overshoot analysis (see Fig. 22 a) the large differences for the thrust between the strain gauge measurement and the reconstructed loads, both with and without the unsteady aerodynamics model, catches the eye. At closer investigation, the dynamic correction based on the estimated eigenfrequency and damping coefficient seems to be unsuited for very detailed comparisons. Therefore, this signal channel is omitted in the discussion of the detailed comparison. For the blade root bending moment and torque, the fit is good.

The 1c model investigation shows a good agreement between the strain gauge measurement derived and reconstruction based $\tau_{single}$ values. They are larger for the step to high load than for the step to low load, which is consistent with what was found before in the analysis of induction and wake results. They have a similar size to the fitted time constants to the axial inductions. The small changes in the overshoot of the reconstructed loads the uA model introduces demonstrate the sensitivity of the time constant fits to these small changes in the signal.

We do assume structural interactions to be the main driver for the observed differences in overshoot and also for the differences in the fitted time constants between the strain gauge signals and reconstructed loads with and without the uA model.

## 5    Conclusions

The objective of the presented dynamic inflow measurements was to deepen the general physical understanding of the dynamic inflow effect for wind turbines.

Direct experimental evidence of dynamic inflow is given through a very clear delay of induction factors at different radial positions at the rotor plane in response to a pitch angle step. Until now, dynamic inflow effects were only proven indirectly through measurements of turbine loads or flow measurements in the wake. We further affirmed that a two time constant model is more suited than a one time constant model to describe the behaviour of the axial induction for such a pitch step. The fast time constant of this model, representing the near wake influence, has a strong radial dependency near the root, however not in the mid and tip region, where it approaches constant values towards the outer part of the blade. The slow time constant, related to the close far wake, shows a slight decrease towards the outer part for the step to low load. The overshooting behaviour of the tangential induction is a novel finding of this work. It could be explained by the shed vorticity, which results from the circulation change during the pitch step. We expect that the radial dependency of the axial induction time constant is related to the observed dynamic wake expansion for the step to low load. With the wake measurement, we affirmed that the formation of the mixed wake after the pitch step convects faster for the step to low load, than for the step to high load. We suppose, that this is the reason for the lower slow time constants of the axial induction for the step to low load. We further found that the mixed wake after the pitch step initially travels at nearly free stream velocity for both pitch directions. We assume that the dynamic inflow effect in this near wake is governed by the shed vortex from the tip due to the fast change in trailed vorticity. These vortices travel at such high velocities. Another finding is that the initial decay of the axial inductions during the pitch step is



similar for both pitch directions. We further identified the aerodynamic characteristics of the turbine to be the reason for the higher load overshoot for the pitch step to low load.

This comprehensive pitch step measurement set allows for detailed validation of engineering models and simulations, as we performed with the induction data in Berger et al. (2020). Further investigations are recommended with high fidelity models of induction effects, eg. FVWM or Actuator Line CFD simulations to support the interpretation of the data. Due to the scaling of induction aerodynamics of this model turbine, based on the NREL 5MW turbine, the general behaviour of the induced velocities and wake are very similar to that turbine. It thus enables non-dimensional comparisons with that reference turbine, a turbine that has been extensively used worldwide in validation studies.

*Data availability.* Preprocessed data will be made publicly available with final publishing.

*Author contributions.* FB designed, performed, processed and analysed the experiment and wrote the manuscript. DO assisted in the experiment, estimated steady corrections and wrote post processing scripts for the LDA data. JGS and MK contributed with several fruitful discussions from an early planning stage on. MK supervised the work. All co-authors thoroughly reviewed the manuscript.

*Competing interests.* The authors declare that they have no conflict of interest.

*Acknowledgements.* This work was partially funded by the Ministry for Science and Culture of Lower Saxony through the funding initiative Niedersächsisches Vorab in the project «ventus efficiens» (Ref. Nr. ZN3024). We thank Iván Herráez for the discussions on the wake induction measurement method. We further thank Dominik Traphan and Tom Wester for their help with the LDA system.



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
