# Peer review of "Experimental analysis of radially resolved dynamic inflow effects due to pitch steps"

_Wind Energy Science, 2021_

## Referee Comment (RC1)

**Experimental analysis of radially resolved**
**dynamic inflow effects due to pitch steps**

The authors present a very interesting, detailed and well designed experimental analysis of the unsteady behavior of the wake-induced inflow in a model-scale wind turbine undergoing step changes in the pitch angle. Moreover, a very important effort is done by the authors to derive some general conclusions about the numerical modelling of dynamic inflow on the basis of their experimental evidences.

The topic of the paper is very interesting and, as the authors state several times, it is very important to have a detailed radial analysis (and not only global) of wake-induced unsteady inflow for the assessment of the available engineering models as well as higher fidelity solvers.

The paper is well written, although there are some minor technical corrections that I have proposed at the end of this document. As a general comment I found the paper quite long. I understand that the presented results are many and they require suitable comments and description. Nevertheless, I encourage the authors to try to make the paper more concise to avoid the reader getting confused by so many details. For example, some of the results could be included in a specific Appendix whilst only the most important ones are retained in the main text.

My indication is to **ACCEPT** the paper only after **MINOR REVISIONS** following the comments listed below.

**SPECIFIC COMMENTS**

1. **Figure 2**: although the description of the experimental setup is very clear, I suggest that the authors include a 3D sketch of it to replace Figure 2. This would greatly help the reader in getting a quick overview of the setup and of the instrumentation.

2. **line 113**: The authors state that the rotor blades are collectively pitched by 5.9°. In Figure 3 the pitch angle before imposing the step seems to be 5°. I suggest to include in the description of the rotor the blade pitch value corresponding to the considered operating conditions.

3. In **Section 2.1.3** the authors make reference to the work by Herràez (2018) for the description of the experimental technique used for the definition of the measurement line (the bisectrix of two rotor blades). For the sake of reading clarity, it would be beneficial to provide a bit more details supporting the choice of that line of measurements by clearly explaining the effect of each blade (bound circulation) and of the shed/trailed vorticity, and providing brief motivations for the limitations of this technique in catching the effect of trailed vorticity (which is very relevant close to tip and root of the blade). In this regard, I suggest also that the rotor plot in figure 4b should be made in 3D in order to better get the information about the point of view and of the direction of circulation on the blades. Finally, in Fig. 4A, the measured velocity components are not clear: is the tangential component a radial one? I believe that a 3D view of the rotor could help also in this regard.

4. If I understood correctly, in **Section 2.1.3** the results of Fig. 5 should be comparable with the data in Herràez 2018. If so, it would be nice to plot the reference numerical data on top of the presented results. Moreover, in the cited paper the induced (perturbation) velocity reaches 0 value at the bisectrix

of each pair of blades, i.e. at azimuth 60°, 180° and 300°. In the present results this is not verified (even if the mean value of the measurement, about 4 m/s, would be eliminated from the data) so the authors are invited to clarify these discrepancies. The operating conditions (wind speed, rpm) are not reported. In the paper by Herràez it is clear that the numerical data are obtained in a phase-locked way, i.e. considering blade 1 at 12 o'clock position and computing the velocity along a circle of radius r. Differently, here the authors seem to consider a fixed point whilst the blade is rotating. Please clarify this point.

5. **Section 2.2:** in the end of the paper the authors correctly state that the proposed methodology for the estimate of the dynamic inflow time constants can be very important for the enhancements of engineering models. From the presented analysis it is also clear that the synthesized time constants depend on several parameter such as the radial position, the pitch direction and (?) TSR. So it would be very interesting to include in the paper a brief discussion on how this methodology could be generalized to be used in an engineering model which must be applied to several load cases in the design of a wind turbine.

6. **Section 2.3**: the authors explain that in the load reconstruction from induction measurements lift and drag coefficients are obtained by Xfoil. Has any correction to take into account for 3D effects been considered? It is well known that sectional loads are typically underestimated if purely 2D polars are used in the framework of BET/BEMT theories. My impression is that loads reconstruction could be improved by the use of 3D-corrected polars. For the NREL 5 MW rotor these are available in the report by NREL describing the turbine characteristics.
In this section it is also mentioned that the model by Pirrung et al. 2017 is used to take into account unsteady airfoil aerodynamics. Even though the paper is cited correctly and easily found in the literature, for the sake of clarity it would be beneficial to add (here or in a devoted Appendix) the main details of this model.
Finally, I did not understand the sentence on line 231 "The typical time lag….": maybe it is related to some aspects of the model by Pirrung? In any case I suggest to rephrase the sentence to clarify the role of the two mentioned time constants.

7. I did not fully understand the sentences from line 255 to 260. My understanding is that the aim is to determine *a priori* a range of variation of the AOA in unsteady (pitch step) conditions. Please rephrase the sentence to make it more clear.

8. From the presented induction results it is evident that, in general, the time constants for model 1c and 2c depend on the considered radial position and on the pitch direction. Moreover, they also depend on the distance of the fitted measured velocity field from the rotor disk. Do the authors have any evidence that they also depend on the TSR? In order to use the 1c and 2c model within an engineering aerodynamic tool, it would be beneficial to have a relationship that somehow links the time constants used for fitting the induction to the operating conditions and the radial station. Moreover, do the author have any proposal on how to generalize the values of the synthesized constants in order to use them in different load cases (different pitch step but not only, for example yawed flow or floating wind turbines)? In other words, do the author think that the time constant values found in this analysis could be used for other turbines in other operating conditions?

9. **Line 306**: the authors highlight that for the tangential induction factor, a different value for t0 was used with respect to the axial case. Moreover, the variation of t0 along the radius is quite relevant. To my understanding, this radial variation was not present in the axial induction. Moreover, a different value of t0 is used also for the fitting of the velocity field further downstream (fig. 17). The author

should comment on this difference and on the sensitivity of the time constants with respect to the choice of t0.

10. **Figure 17**: might the drop in velocity at 0.2 R before the new equilibrium somehow be explained by the effect of the nacelle? Which is the radius of the nacelle?

11. **Figure 19:** The caption is not clear (as well as the text on line 368). In particular I did not understand the definition of the wake ramp. Maybe a sketch could help in this regard.

12. **Section 3.3**: the paper includes several results. I don't think that in this section the (SG-no corr) results are really necessary as all the other ways of computing loads presented in this section do not include the mentioned correction.

13. **Line 390-391**: I did not fully understand this statement.

14. **Line 394-395**: as already pointed out in my previous comments, might the use of the 2D polars instead of the 3D corrected one form an explanation for the deviation of LDA recon loads with respect to the strain gauges measurements? Moreover, at the end of the discussion the authors state that the main driver for the observed differences between reconstructed loads and SG measurements are the structural interactions. I suggest to investigate also the effect of 3D flow phenomena that are not fully included when using purely 2D polars.

15. **Line 408-416:** the theoretical procedure described in these lines to obtained the results in Fig. 23 is not clear to me. Please rephrase the paragraph for the sake of clarity.

16. **Line 453**: it would be nice (not only here but in general in the discussion of the results) to indicate the percentage radial variation of t_slow and t_fast because the have very different magnitude and from the plot is not immediately evident.

**TECHNICAL CORRECTIONS**

1. Throughout the paper large use of personal forms (like "we", "us", "our"…) is made. I find the impersonal forms to be more appropriate for a scientific paper. Please revise the whole manuscript taking care of this aspect.

2. In the abstract and in the introduction both present and past tenses are used when referring to literature results and also presented results: please make a coherent choice. I would suggest to use always the present tense.

3. line 84: pitching speed should be indicated in rad/s

4. line 130: "blades induction"

5. line 236: replace "along" with "with" and "from" with "described in".

6. line 248: "low load case shows"

7. line 292: replace "shorter" with "smaller"

8. line 299: symbol t_fit was never defined before this point

9. line 525: replace "adopt" with "adapt"

---

## Author Comment (AC1)

**"Experimental analysis of radially resolved dynamic inflow effects due to pitch steps"**
Frederik Berger, David Onnen, J. Gerard Schepers, and Martin Kühn,
**Author response to reviewer comments**

We would like to thank Luca Greco and Georg Raimund Pirrung for their thorough review, time and constructive and very meaningful comments. Their input helped to improve the original manuscript.

We addressed all comments and reply to these point by point. First the comment is repeated (in italics), followed by an answer of the authors and if applicable the excerpt from the LaTeX-Diff file (framed), highlighting the changes. Line numbers in the comments refer to the discussion version and line numbers in the response to the LaTeX-Diff of the revised manuscript, which is also attached as a complete version.

**Luca Greco, Reviewer #1**

**Reviewer #1 general comment:**

0. **[Reviewer #1]** *As a general comment I found the paper quite long. I understand that the presented results are many and they require suitable comments and description. Nevertheless, I encourage the authors to try to make the paper more concise to avoid the reader getting confused by so many details. For example, some of the results could be included in a specific Appendix whilst only the most important ones are retained in the main text.*

   **[Authors]** We completely see the point here. We approached this such that we identified the major messages in our conclusions and which measurements respectively analysis are needed for our argumentation to get to these conclusions. We identified the time constant analysis of the hot wire signals to be not relevant to reach our conclusions and thus moved this part to the appendix as suggested.

**Reviewer #1 specific comments:**

1. **[Reviewer #1] Figure 2**: *although the description of the experimental setup is very clear, I suggest that the authors include a 3D sketch of it to replace Figure 2. This would greatly help the reader in getting a quick overview of the setup and of the instrumentation.*

   **[Authors]** Thank you for this comment. We added another perspective (top view) of the setup for an improved quick overview of the setup. With this view the position of the LDA probe head on the traverse is obvious as well as the spread of the measurement positions of the hot wire and LDA focus points. We considered a 3D CAD derived sketch, however stayed with a 2D representation and added a second perspective view as we found it more clear and easier/quicker to see the coordinate system orientation. See below the updated Fig. 2.

[Figure]

**Figure 2.** Sketch of the setup in the wind tunnel. View from (a) the side and (b) the top.

2. **[Reviewer #1] line 113**: *The authors state that the rotor blades are collectively pitched by 5.9°. In Figure 3 the pitch angle before imposing the step seems to be 5°. I suggest to include in the description of the rotor the blade pitch value corresponding to the considered operating conditions.*

   **[Authors]** We added the actual pitch settings as suggested in the text (see line 117)

   > The rotor blades are collectively pitched by $5.9°$ between $-0.9°$ and $5.0°$, within $0.070\,\text{s}$, corresponding to about half a rotor revolution and half the reference time. The pitch step is from a low rotor load at a thrust coefficient $C_T = 0.48$ to a high load at $C_T = 0.90$ and vice versa, based on the strain gauge derived thrust. This corresponds to rotor effective inductions of
   > 120 $a_{eff} = 0.14$, respectively $a_{eff} = 0.34$, based on the momentum theory relation ($C_T = 4a(1-a)$).

3. **[Reviewer #1]** In **Section 2.1.3** *the authors make reference to the work by Herràez (2018) for the description of the experimental technique used for the definition of the measurement line (the bisectrix of two rotor blades). For the sake of reading clarity, it would be beneficial to provide a bit more details supporting the choice of that line of measurements by clearly explaining the effect of each blade (bound circulation) and of the shed/trailed vorticity, and providing brief motivations for the limitations of this technique in catching the effect of trailed vorticity (which is very relevant close to tip and root of the blade). In this regard, I suggest also that the rotor plot in figure 4b should be made in 3D in order to better get the information about the point of view and of the direction of circulation on the blades. Finally, in Fig. 4A, the measured velocity components are not clear: is the tangential component a radial one? I believe that a 3D view of the rotor could help also in this regard.*

   **[Authors]** This is a very helpful comment. The line of measurement is mainly defined by constraints from the LDA head mount options and the effect of the tower. We also elaborated further on the effect of each blade and the bound and trailed vorticity in the text. Figure 4 and caption was also modified to help understand the general concept quickly. In Fig. 4b firstly a coordinate system is introduced, and the orientation of the rotor is described in the caption. The vorticity circles are also modified according to the perspective to show the explicit direction. The vector orientations in Fig. 4a are also mentioned in the caption in terms of the coordinate system. The text additions (lines 139 - 149) and the revised Fig. 4 are shown below.

In Fig. 4 a the MoWiTO turbine is shown with the LDA laser beams and the probed axial ($u_{ax}$) and tangential ($u_{ta}$) velocity components at a specific radius. Alongside in Fig. 4 b, the concept of the counterbalancing of the bound circulation of the evenly loaded blades is sketched.  For the shown position the blade at the 9'o clock position has no influence on

140  the axial velocity on the indicated line of measurements. At that line the downwash of the blade ahead of the indicated line counteracts the upwash of the blade behind it and they cancel each other. The velocity at the line thus is only influenced by the wake induction. Herráez et al. (2018) argue, that the trailed vorticity especially at the tip might play a non-negligible role, as it cannot be captured well at the high distance between the measurement position and the blade tip. Therefore the method is less suited for the root and tip region of the blade.

145  Two constraints defined the line of LDA measurements. The first is the height range of the LDA probe head, which is from tower bottom to hub height. The second is to minimize the influence of the tower on the blade nearest to the tower for the measurements in the bisectrix of two blades. This led to a measurement line at the 3'o clock position. The tower does disturb the axial symmetry, however, based on an estimation of the tower effect with a dipole model as in Schepers (2012) the tower effect  on the 5'o clock blade position is considered negligible.

[Figure]

Figure 4. (a)  Clockwise rotating MoWiTO 1.8 with 2D-LDA probing axial (along x-axis) and tangential velocity (along z-axis) components in the bisectrix of two blades. (b) Scheme of  counterbalancing bound circulation of the evenly loaded blades  of front view of clockwise rotating rotor. The LDA measurement line is indicated in  green (adapted from Herráez et al. (2018)).

4.  *[Reviewer #1] If I understood correctly, in **Section 2.1.3** the results of Fig. 5 should be comparable with the data in Herràez 2018. If so, it would be nice to plot the reference numerical data on top of the presented results. Moreover, in the cited paper the induced (perturbation) velocity reaches 0 value at the bisectrix of each pair of blades, i.e. at azimuth 60°, 180° and 300°. In the present results this is not verified (even if the mean value of the measurement, about 4 m/s, would be eliminated from the data) so the authors are invited to clarify these discrepancies. The operating conditions (wind speed, rpm) are not reported. In the paper by Herràez it is clear that the numerical data are obtained in a phase-locked way, i.e. considering blade 1 at 12 o'clock position and computing the velocity along a circle of radius r. Differently, here the authors seem to consider a fixed point whilst the blade is rotating. Please clarify this point.*

**[Authors]** This is a good point and we totally agree that our original  statement was far fetched as you point out. We followed your suggestion and added the analytical solution according to the derivation by Herraez et al. 2018 to Fig. 5a (see below). The implementation and also the differences to the case presented in the Herraez et al. 2018 paper are outlined in Appendix A (lines 655-670). We further added the info that the shown measurement is from the steady high load operational point, that is defined priorly (line 151).

150    To obtain the values in the bisectrix,  the LDA system is synchronised with the MoWiTO data acquisition system. Measurements at  the constant high load are plotted for one position of the axial and tangential probe over the azimuth angle $\phi_1$ of the turbine in Fig. 5. The bisectrix values that are in a threshold of $\pm 3°$ are marked in red.

155    In Fig. 5 a also the analytical course seen by the axial probe according to Herráez et al. (2018) (see App. A), is presented and shows a good match to the measured signal. This good fit gives a high level of confidence on the applicability of the method to this experiment. For the tangential probe, data is missing around $-1.3\,\mathrm{ms}^{-1}$ and also at $2.3\,\mathrm{ms}^{-1}$ for the axial probe, which

[Figure]

**Figure 5.** (a) Measurements of axial probe for high load case at a radius of $0.7\,R$ for 400 revolutions over azimuth angle $\phi_1$ with marked data within the bisectrix threshold. $\phi_1 = 0°$ relates to the 12 o'clock position of blade 1. (b) Analogously for the tangential probe.

655    **Appendix A:  Counterbalance of the blade induction at the bisectrix between consecutive blades**

The detailed derivation based on the theorem of Biot-Savart is given in Herráez et al. (2018) but the relevant formulas to produce the analytical solution in Fig. 5 a are given here.

Eq. (A11) of Herráez et al. (2018) is reproduced in Eq. (A1). The velocity induced by each of the $N$ rotor blades (numbered $i \in [1, 2, .., N]$) is given in dependance of the azimuth angle of the blade $\psi_i$, the azimuth angle of the sampling point $\beta$ and the

660  coordinates along the blade spanwise direction $l$. The circulation along the blade span is given by $\Gamma_i(l)$ and the azimuth angle of all blades can be defined by $\psi_i = \psi_1 + (i-1)\Delta\psi$, where $\Delta\psi = \frac{2\pi}{N}$.

$$u_i(r) = \frac{1}{4\pi} \int_0^R \Gamma_i(l) \frac{r \sin(\psi_i - \beta)}{[r^2 + l^2 - 2lr\cos(\psi_i - \beta)]^{\frac{3}{2}}} \,\mathrm{d}l \tag{A1}$$

Herráez et al. (2018) use a phase lock of the rotor for the simulations and compute the velocity along a circle for a specific radius. For the PIV validation in Herráez et al. (2018) and the application in this paper a stationary probe is used and the

665  rotor position is varied. The circulation $\Gamma_i(l)$ is obtained from the lift distribution calculated in Eq. 9 by the Kutta–Joukowski theorem.

The analytical solution of the axial probe $u_{probe,analytical}$ in Fig. 5 a considers an axial velocity of $u_{ax} = 4ms^{-1}$ and the combined effect of all three blades according to Eq. (A2). Note that the Eq. (A1) has a singularity for the probe position and a blade element being at the exact same position, that are excluded from the plot ($90°$, $210°$ and $330°$).

670
$$u_{probe,analytical}(r) = \sum_{i=1}^{N} u_i(r) + u_{ax}(r) \tag{A2}$$

5. *[Reviewer #1] Section 2.2: in the end of the paper the authors correctly state that the proposed methodology for the estimate of the dynamic inflow time constants can be very important for the enhancements of engineering models. From the presented analysis it is also clear that the synthesized time constants depend on several parameter such as the radial position, the pitch direction and (?) TSR. So it would be very interesting to include in the paper a brief discussion on how this methodology could be generalized to be used in an engineering model which must be applied to several load cases in the design of a wind turbine.*

    **[Authors]** That is a good point. We added a paragraph at the end of Sect. 2.2 describing how the so obtained time constants can be used to assess the time constants of the Øye and new DTU dynamic inflow model and what the typical scaling parameters to size and operation are (lines 231-235).

> These time constants can be used for comparison and tuning of the time constants in the dynamic inflow engineering models of Øye (see Snel and Schepers (1995)), used in OpenFAST and GH Bladed, and the new DTU model (see Madsen et al. (2020)), used in HAWC2. The time constants of these engineering models are derived from simulations and parametrised to the turbine size and operational condition considering the relevant dynamic inflow time scaling factor $\frac{u_0}{R}$, the radial position and
> 235 the quasi-steady axial induction factor.

    Using these three common scaling quantities we get time constants for two axial induction settings, that give a good span for the relevant induction settings and such are a good basis for validation of existing models (see also Berger et al. 2020 https://doi.org/10.1088/1742-6596/1618/5/052055). They are however not sufficient for development of new models. We added two sentences in the conclusions on a further planned experiment to increase this data basis and enable model development from experimental data (line645-650).

> Further investigations are recommended with  high-fidelity models of induction effects, eg. FVWM or Actuator
> 645 Line CFD simulations to support the interpretation of the data  and improvement of models. Further planned steps to extend the understanding of the dynamic inflow process and enhance or develop models on the basis of this experimental setup are the generation of a wider database of pitch steps with varying parameters of inflow velocities, rotor speeds and rotor induction levels but also more realistic inflow conditions, including e.g. non-uniform inflow and gusts. With a wider database it can be tested if besides the axial induction, radial position and typical dynamic inflow time scaling factor $\frac{u_0}{R}$ further parameters have
> 650 a significant influence on the time constants, as e.g. the operating TSR, background turbulence or non-uniform inflow. Due to

6. *a [Reviewer #1] In Section 2.3 the authors explain that in the load reconstruction from induction measurements lift and drag coefficients are obtained by Xfoil. Has any correction to take into account for 3D effects been considered? It is well known that sectional loads are typically underestimated if purely 2D polars are used in the framework of BET/BEMT theories. My impression is that loads reconstruction could be improved by the use of 3D-corrected polars. For the NREL 5 MW rotor these are available in the report by NREL describing the turbine characteristics.*

    **[Authors]** That is a very good point we originally did not consider. We implemented the quiet common model by Snel (line 250-252) on the lift polar to account for 3D effects. There is a slight effect of higher reconstructed thrust at the high load operational point reducing the mismatch to the strain gauge measured from -7% to -6%. The other load channels and the loads at the low load configuration stay constant.

> 250 by XFoil (see Drela (1989)). The  lift polars are corrected for 3D effects, as lift coefficients on rotating blades can be significantly higher than for a stationary blade due to the effect of cross lows related to a stall delay. This effect is most relevant for the root airfoils and was corrected by the method by Snel et al. (1993). The blade segment width is $\Delta r$ and $c$ the chord

*b [Reviewer #1] In this section it is also mentioned that the model by Pirrung et al. 2017 is used to take into account unsteady airfoil aerodynamics. Even though the paper is cited correctly and easily found in the literature, for the sake of clarity it would be beneficial to add (here or in a devoted Appendix) the main details of this model. Finally, I did not understand the sentence on line 231 "The typical time lag....": maybe it is related to some aspects of the model by Pirrung? In any case I suggest to rephrase the sentence to clarify the role of the two mentioned time constants.*

[Authors] Thank you for this helpful comment. We see the point, that some more information on the uA model would be helpful already in this paper. Thus we reproduced the main model in Appendix B (lines 671-679).

We further clarified the sentence on the role of the two time constants (line 260-261)
* * *
**Appendix B: Unsteady aerodynamics model**

The uA model as described in detail in Pirrung et al. (2017) Sect. 2.3 is reproduced with slightly altered variable names. The model consists of two filter functions Eq. (B2, B3), which use the same time constant $\tau_{uA}^j$, defined in Eq. (B1). The index $j$ denotes the time step and the index QS the quasi-steady solution. The time step of the simulation, respectively reconstruction,
675    is defined by $\Delta t$. The effective angle of attack $\alpha_{eff}^j$, that indcludes the uA effect, is obtained by Eq. (B4).

$$\tau_{uA}^j = \frac{c}{2u_{rel}^j} \tag{B1}$$

$$x_1^j = x_1^{j-1} \exp\left(-0.0455\frac{\Delta t}{\tau_{uA}^j}\right) + \frac{1}{2}\left(\alpha_{QS}^j + \alpha_{QS}^{j-1}\right) 0.165 u_{rel}^j \left(1 - \exp\left(-0.0455\frac{\Delta t}{\tau_{uA}^j}\right)\right) \tag{B2}$$

$$x_2^j = x_2^{j-1} \exp\left(-0.3\frac{\Delta t}{\tau_{uA}^j}\right) + \frac{1}{2}\left(\alpha_{QS}^j + \alpha_{QS}^{j-1}\right) 0.335 u_{rel}^j \left(1 - \exp\left(-0.3\frac{\Delta t}{\tau_{uA}^j}\right)\right) \tag{B3}$$

$$\alpha_{eff}^j = \frac{1}{2}\alpha_{QS}^j + \left(x_1^j + x_2^j\right)/u_{rel}^j \tag{B4}$$
* * *
260    influenced. The  main model is reproduced in App. B. The typical uA time constant which determines the lag of the angle of attack due to the uA effect is in the order of  $\frac{c}{u_{rel}}$, whereas the typical time constant of the dynamic inflow effect is  $\frac{u_0}{R}$ and at least two magnitudes of size larger, as mentioned in Sect. 1. Reconstructed loads will be investigated with and without the uA model.
* * *
7.   *[Reviewer #1] I did not fully understand the sentences from line 255 to 260. My understanding is that the aim is to determine a priori a range of variation of the AOA in unsteady (pitch step) conditions. Please rephrase the sentence to make it more clear.*

[Authors] Your understanding is correct. We added a sentence (line 286-287) to state the reason and give better context and tried to further improve the readability of the whole paragraph (line 284-292).

The observed difference between the two angle of attack distributions is smaller than the pitch step value of 5.9° the blades
285  do pitch, as . This is due to the flow through the rotor and induction factors that change between the two steady operational
states. With these steady levels, the From this information the maximum angle of attack range can be estimated a priori, in
order to assess the flow conditions during the pitch steps. The dynamic maximum and minimum angle of attack distributions
can be are estimated for an infinitely fast pitch step, only considering the influence of the wake. For this, we . We assume in a
mind experiment that the flow field of the old steady state steady state before the pitch step is unchanged, but the pitch step and
290  thus geometrical change of the inflow angle is already done, giving us . This allows to estimate the extreme dynamic angles of
attack. The flow field adapts to the new equilibrium and the new steady level just after the , neglecting any damping uA effects
for the infinitely fast pitch step is terminated.

8. **[Reviewer #1]** *From the presented induction results it is evident that, in general, the time constants for model 1c and 2c depend on the considered radial position and on the pitch direction. Moreover, they also depend on the distance of the fitted measured velocity field from the rotor disk. Do the authors have any evidence that they also depend on the TSR? In order to use the 1c and 2c model within an engineering aerodynamic tool, it would be beneficial to have a relationship that somehow links the time constants used for fitting the induction to the operating conditions and the radial station. Moreover, do the author have any proposal on how to generalize the values of the synthesized constants in order to use them in different load cases (different pitch step but not only, for example yawed flow or floating wind turbines)? In other words, do the author think that the time constant values found in this analysis could be used for other turbines in other operating conditions?*

**[Authors]** This actually is a quite interesting comment. For this experimental campaign we only considered one TSR and two thrust levels. On that basis we thus cannot comment on a dependence on TSR. Until now the general understanding in the community is that the dynamic inflow effect depends on the thrust coefficient, respectively axial induction, the ratio of wind to radius ($R/u_0$) as well as radial position. In our view for a generalization only based on these measurements a wider range of pitch steps and conditions is needed. We emphasized that point at the end of the conclusions in an outlook (lines 642-647; Q5 above) and added these points to our list for future experiments on dynamic inflow. An existing model like the Øye model however, that uses the mentioned scaling parameters can be tuned based on these two relevant thrust settings.

9. **[Reviewer #1] Line 306**: *the authors highlight that for the tangential induction factor, a different value for t0 was used with respect to the axial case. Moreover, the variation of t0 along the radius is quite relevant. To my understanding, this radial variation was not present in the axial induction. Moreover, a different value of t0 is used also for the fitting of the velocity field further downstream (fig. 17). The author should comment on this difference and on the sensitivity of the time constants with respect to the choice of t0.*

**[Authors]** This is a valid point. For quantities with an overshoot, like the tangential induction such changes might have a relevant effect on the fitted time constants. For this specific case we ran the fitting again with the strictly fixed $t_0$ to the end of the pitch step and compared the results. This led to a reduction in the number of valid fits by 1/3, however for the rest of the measurement points similar results were obtained. We added this in the text (line 351-354) and also Fig. 14.
For the hot wire time constant fit (that was moved to the Appendix C) we also sharpened the text to better explain the approach to obtain the start point of the time constant fit (line 681-685).
For the axial induction that does not show an overshoot we found no relevant effect of such small changes in the starting point of the fit.

[Figure]

**Figure 14.** One time constant model fit of $\tau_{single}$ to the tangential wake induction over the radius for both pitch directions.

The influence of the small allowed shift in the start of the fit is also investigated. For a strict starting point of the fit at $t_0$, shown in Fig. 14 as stars, the set criterion of overshoot to noise is fulfilled for only three radii for the step to high load and five radii for the step to low load. For the available radii however there are only negligible differences to the shown fit in comparison to the observed spread.

355 **3.2 Wake results**

The 1c model fit is applied to the hot wire signal shown in Fig. 17. The fit does start at $t_0$, which is defined at the point where the velocity has adjusted by $28\%$ to the new equilibrium. This definition is less accurate than the end of the pitch step, as for the axial induction transients. However no direct fitting start time can be defined and this value is based on the mean value the axial inductions have adjusted during the pitch step. The aim of this definition is to make the time constants comparable to 685 these time constants of the axial wake induction.

10. *[Reviewer #1] Figure 17: might the drop in velocity at 0.2 R before the new equilibrium somehow be explained by the effect of the nacelle? Which is the radius of the nacelle?*

**[Authors]** The nacelle has a radius of 0.1R. We do not expect a relevant contribution from the nacelle wake at 0.2R (line 391-392).

both shown distances. The effect of the nacelle is assessed as unlikely to be the reason for this structure, as the nacelle only has a radius of $0.1\,R$. At the radius of $0.6\,R$ the signal decays exponentially. At 1 R the signal increases quickly to a peak at nearly free stream velocity before decreasing exponentially to the new steady value. The signals at $0.5\,D$ behind the turbine for the 395 two higher radii, $0.6\,R$ and $1\,R$, show a better fit with the exponential decay function. For the farther distance at $1.5\,D$ the hot

11. *[Reviewer #1] Figure 19: The caption is not clear (as well as the text on line 368). In particular I did not understand the definition of the wake ramp. Maybe a sketch could help in this regard.*

**[Authors]** Thank you for pointing out this unclear description. In a first step we changed the name of the wake ramp to wake front, for the analogy of a weather front. We changed the text for the introduction extensively as shown below. The general definition is sharpened in lines 389-390. The approach is reworked and extended in lines 406-421. The shortened caption is also shown (note that it is now Fig 18 due to a rearrangement before in the text and therefore the LaTeX-Diff highlighting did not work here, indicating everything in the caption as new).

For the radius at 0.6 R a vertical line, called  a wake front, marks a characteristic $t_1$ of the exponential
390 decay, where the signal has adapted by $50\%$ to the new equilibrium value. For the radius of $0.2\,R$ the velocity transient shows

In the next step,  the wake front velocity is analysed to measure how fast the transition
point (the wake front) between the old wake and the new wake
 convects. This wake front can be thought of as being similar to a weather front. We define the wake front velocity by
the time this characteristic wake front needs to travel from one considered downstream position to the next. So exemplarily
410 in Fig. 17 at $0.6\,R$ there is the wake front for $0.5\,D$ at $t_{ramp,0.5D} = 0.20\,s$ $t_{front,0.5D} = 0.20\,s$ and
for $1.5\,D$ at $t_{ramp,1.5D} = 0.56\,s$ $t_{front,1.5D} = 0.56\,s$. Within this time difference, the wake front thus has travelled by $1\,D$,
giving a mean wake front velocity between these downstream distances.

In Fig. 18, this wake front velocity is shown for both pitch directions, normalised by the free stream velocity.

415

 The wake front
velocity is obtained by considering a mean value of the hot wire positions at the radii of $0.4\,R$, $0.6\,R$ and $0.8\,R$
. These signals were weighted based on their position with the
conservation of mass in mind. For that each position was attributed an annulus reaching $\pm\,0.1\,R$
420 from the radius of the position. This mean velocity signal at the different downstream distances, which is used to obtain the
wake front velocity, represents in this definition ($0.3\,R$ to $0.9\,R$) the major part of the swept area of the rotor. Thus between

[Figure]

**Figure 18.** Velocity of the wake front for both pitch directions, normalised by the free wind velocity. For orientation also the theoretical
normalised wind velocity in the rotor plane $(1-a)$ and in the far wake $(1-2a)$, based on the measured thrust coefficient, are indicated for
both steady states.

12. ***[Reviewer #1] Section 3.3****: the paper includes several results. I don't think that in this section
    the (SG-no corr) results are really necessary as all the other ways of computing loads
    presented in this section do not include the mentioned correction.*

    **[Authors]** In the context the figure is described we completely agree and consequently
    removed that line from the plot as suggested.

13. ***[Reviewer #1] Line 390-391****: I did not fully understand this statement.*

    **[Authors]** We reworked the formulation of that sentence to make the statement more clear
    (line 445-448).

445 The slight overprediction of reconstructed loads for the $M_{flap}$ and $M_{aero}$ on the one hand and the underprediction of $F_{thrust}$
for is attributed to a higher influence of the larger radii. For the first two signals
 the blade acts as a lever arm for the sectional forces, thus giving them a higher weighting. In contrast the sectional
forces are added without considering a lever arm for $F_{thrust}$.

14. ***[Reviewer #1] Line 394-395**: as already pointed out in my previous comments, might the use of the 2D polars instead of the 3D corrected one form an explanation for the deviation of LDA recon loads with respect to the strain gauges measurements? Moreover, at the end of the discussion the authors state that the main driver for the observed differences between reconstructed loads and SG measurements are the structural interactions. I suggest to investigate also the effect of 3D flow phenomena that are not fully included when using purely 2D polars.*

**[Authors]** As also put in the answer to question 6a in slightly more detail we now consider 3D effects for the lift polars and achieved a slight improvement for the reconstructed thrust at the high load level.

The observed differences you refer to in the text are the dynamic differences, so mainly the overshoot and time constant. Here especially the mismatch in the thrust overshoot can be related to the oscillation of the tower after the pitch step, that we could not completely filter out. For future experiments we will consider an accelerometer in the nacelle to consider these inertial forces, however the necessity of such a sensor is one of the lessons learned from this experimental campaign. Also the flexibility of the blades (very stiff but not perfectly stiff) and drivetrain add uncertainty to these highly dynamic experimental measurement.

15. ***[Reviewer #1] Line 408-416**: the theoretical procedure described in these lines to obtained the results in Fig. 23 is not clear to me. Please rephrase the paragraph for the sake of clarity.*

**[Authors]** Thank you for this hint. We rephrased that paragraph to increase the clarity. The reworked paragraph is shown below in (lines 465-475)

465  In order to assess the reasons for the differences in load overshoot it is of interest to investigate the theoretical maximum load overshoot when changing from one operational point to the other. This can be estimated based on the steady  states of the operational points and the theoretical dynamic maximum and minimum angle of attack distribution, as was  introduced in Fig. 8 . The steady integral loads for high and low load are reconstructed as before for Fig. 19.  For the

470 maximum and minimum load the maximum and minimum dynamic angle of attack  distributions are used. The inflow to the segments is however defined by the inductions for the steady state just before the pitch step . An infinitely fast pitch step is assumed , where the geometrical change of the blade pitch already happened, but the flow did not start to adapt. This apporach neglects uA effects that showed to reduce the overshoot . The derived theoretical maximum amount of overshoot based on the difference of maximum, respectively minimum load and the following

475 steady load is plotted in Fig. 22. A clear difference can be seen between the pitch directions for the axial loads $M_{\text{flap}}$ and

16. ***[Reviewer #1] Line 453**: it would be nice (not only here but in general in the discussion of the results) to indicate the percentage radial variation of t_slow and t_fast because the have very different magnitude and from the plot is not immediately evident.*

**[Authors]** We implemented that good point within the results (line 324-325 & 328-330) and discussion (line 514-517) section. Looking at these values we also slightly updated the conclusions (line 629-631).

The fitted $\tau_{fast}$ is high near the root for both pitch directions. For the step to high load $\tau_{fast}$ decreases from the root to $0.4\,R$, after which there is  an increase again (ignoring an outlier at $0.8\,R$). From $0.5\,R$ to $0.9\,R$ $\tau_{fast}$ increases by 67%.

325 For the step to low load $\tau_{fast}$ has  more constant values from $0.5\,R$ to $0.9\,R$, decreasing by 14%. Hence $\tau_{fast}$ is  smaller for the negative load step, for radii larger than $0.5\,R$, which represents 75% of the rotor swept area.

Values of $\tau_{slow}$ are slightly higher for the step to high load and show more variation than in the prior fit with $k = k_{\text{free}}$ ratio. There is a slight radial trend to higher values, with an increase in $\tau_{slow}$ from $0.3\,R$ to $0.9\,R$ by 11%. For the step to low load also more variation is apparent and a slight radial trend towards lower values is indicated, with a decrease in $\tau_{slow}$ from $0.3\,R$

330 to $0.9\,R$ by 16%. Taking the mean value over radius, the slow time constant for the step to low load is about 28% lower.
* * *
510 VI experiment, Sørensen and Madsen (2006) also saw this high $\tau_{fast}$ values at the root nearest radius of $0.3\,R$ for the step to low load, but not for the step to high load. In  the present and the phase VI experiment, this high $\tau_{fast}$ near the root is of the order of $\tau_{slow}$. In  the present experiment, there is a  difference in the $\tau_{fast}$ between pitch directions, with lower values for the step to low load from $0.5\,R$ to the tip. This difference between the pitch directions is not seen by Sørensen and Madsen (2006). Furthermore a radial dependance can also be seen from the mid of the blade towards the tip with an increase

515 for the step to high load (67%) and a slight decrease for the step to low load ($-14\%$).

For $\tau_{slow}$, a very slight radial dependency can be seen for the fit with $k_{fix}$, with slightly increasing values (11%) towards the tip for the step to high load and slightly decreasing values ($-16\%$) for the step to low load, similar to the fit of $\tau_{single}$.
* * *
model is more suited than a one time constant model to describe the behaviour of the axial induction for such a pitch step. The fast time constant of this model, representing the near wake influence, has a strong radial dependency near the root,  and clear respectively slight dependency in the mid and tip region

630  for the step to high, respectively low load. The slow time constant, related to the close far wake, shows a slight decrease towards the outer part for the step to low load and a slight increase for the step to high load. The overshooting
* * *
**Reviewer #1 technical corrections:**

17. *[Reviewer #1] Throughout the paper large use of personal forms (like "we", "us", "our"...) is made. I find the impersonal forms to be more appropriate for a scientific paper. Please revise the whole manuscript taking care of this aspect.*

    **[Authors]** We changed the vast majority of these points to an impersonal form and only left this personal form at single instances where an opinion or presumption was stated.

18. *[Reviewer #1] In the abstract and in the introduction both present and past tenses are used when referring to literature results and also presented results: please make a coherent choice. I would suggest to use always the present tense.*

    **[Authors]** Thank you for pointing this out. We changed all to present tense as suggested.

19. *[Reviewer #1] line 84: pitching speed should be indicated in rad/s*

    **[Authors]** We added the pitching speed in rad/s.

20. *[Reviewer #1] line 130: "blades induction"*

    **[Authors]** We changed this according to the suggestion.

21. *[Reviewer #1] line 236: replace "along" with "with" and "from" with "described in".*

[**Authors**] We changed this according to the suggestion.

22. [***Reviewer #1***] *line 248: "low load case show**s**"*

    [**Authors**] We changed this according to the suggestion.

23. [***Reviewer #1***] *line 292: replace "shorter" with "smaller"*

    [**Authors**] We changed this according to the suggestion.

24. [***Reviewer #1***] *line 299: symbol t_fit was never defined before this point*

    [**Authors**] We looked this up and actually the variable $t_{fit}$ is introduced in line 205 along the method of the fitting procedure in Sect. 2.2.

25. [***Reviewer #1***] *line 525: replace "adopt" with "adapt"*

    [**Authors**] We changed this according to the suggestion.

**Georg Raimund Pirrung, Reviewer #2:**

**Reviewer #2 minor comments:**

26. [***Reviewer #2***] *I think the angle 'theta' is not consistent between Equation (3) and Equations (11) and (12). In Equation (3) it is the sum of twist and pitch, and in Equations 11 and 12 it seems to be the inflow angle.*

    [**Authors**] Thank you for pointing this out to us. You are completely right. We changed the variable in Eq. (3) for the sum of pitch and twist to 'gamma'.

27. [***Reviewer #2***] *Figure 11: I suggest to remove the irrelevant time constant tau_fast where k_free=1*

    [**Authors**] That is a good idea. We implemented it according to your suggestion in the Fig. 11 and mentioned it in the text (line 312)

> 310    In Fig. 11, the three fitting parameters of the 2c model are presented. In the top row, the three fitted variables $k = k_{\text{free}}, \tau_{fast}$
> and $\tau_{slow}$ are plotted over the radius. Near the root at $0.25\,R$ the $k$ value for both pitch directions has a value of 1 respectively
> nearly 1, indicating no contribution from $\tau_{fast}$, which consequently is also not plotted for these radii. For radii up to $0.5\,R$

28. [***Reviewer #2***] *Figure 16: It could be made a bit more clear what is actually shown in the figure. If I understand it correctly it is abs((u-u_0)/u_0)*

    [**Authors**] Thank you for this comment. We added a sentence to clarify this point (line 373-375).

>    steady state in Fig. 16. The normalised difference is defined as $\Delta u / u_0$, with $\Delta u = u_t - u_\infty$, where $u_t$ is the velocity at the
> respective time at a measurement position and $u_\infty$ the velocity at the same point at the new steady condition after the pitch
> 375   step. Therefore, a value of 0.5 means that the wake has to adapt by $0.5\,u_0$ to reach the new equilibrium. The starting conditions

29. **[Reviewer #2]** *Page 22 line 395 'High deviations are seen for Fthrust between strain gauge measurement and reconstructed loads, especially with the uA model, but also without the model, for both pitch directions.' Was the airfoil data 3D corrected? Without 3D correction the aerodynamics at the inboard sections might be inaccurately predicted by the load reconstruction procedure. This effect would be less visible on flapwise blade root moment and torque due to the short moment arms at the root section.*

**[Authors]** That is a good point. As also written in the answers to question 6a and 14 of the first reviewer we originally did not consider 3D corrected polars. Now we corrected our lift polars with the model proposed by Snel. We do achieve a slightly better match of the thrust at the high load case as you assumed. However there still is some relevant mismatch. We suspect that also the low Reynolds number polars with XFoil (especially those at the root which are at Re 60e3 at 0.2R; The airfoil is a low Re airfoil with low camber, but with 16% relative thickness rather thick for a model turbine) have relevant uncertainties and might lead to a further underprediction of the actual lift forces. Along this line also laminar separation bubbles at this root near stations, that can increase the local lift significantly, are possible. Such a reattaching bubble near the leading edge at the suction side is indicated by a high level of vorticity at a similar operational point at 0.25 R in a PIV measurement of the exact turbine with these blades for an angle of attack of 13° (published only in a presentation https://zenodo.org/record/3955740#.YURSMS221pQ slide 26 top left). We have added this suspicion to the text (line 585-587).

> For the steady equilibrium, the comparison between the strain gauge measurements and the reconstructed loads (see Fig. 20) yields in generally good agreement, apart from the thrust at low load,
> 585 . Especially the here relevant high angle of attack region for the root airfoil is suspected to have uncertainties related to the low Reynolds number and possible laminar separation bubbles that can significantly increase the local lift.

**Reviewer #2 small comments:**

30. **[Reviewer #2]** *'tower bottom' bending moment is probably more frequently used than 'tower foot'*

**[Authors]** We implemented the change as suggested throughout the manuscript.

31. **[Reviewer #2]** *Page 10 line 230 'as a time lag on the angle of attack alpha'. You might add 'and has been extended to take the effect of camber into account'.*

**[Authors]** We implemented the addition also considering a reformulation suggested in question 6b (line 257-259)

> given in detail in Pirrung et al. (2017) is used. This is the inviscid part of the unsteady aerodynamics model by Hansen et al. (2004), which treats the shed vorticity effects due to fast angle of attack changes as a time lag on the angle of attack $\alpha$ and

> has been extended to take the effect of camber into account. Thus, the magnitude and direction of the aerodynamic forces are
> 260 influenced. The  main model is reproduced in App. B. The typical uA time constant which determines the lag of the angle of attack due to the uA effect is in the order of $c/u_{rel}$, whereas the typical time constant of the dynamic inflow effect is $R/u_0$ and at least two magnitudes of size larger, as mentioned in Sect. 1. Reconstructed loads will be investigated with and without the uA model.

32. *[Reviewer #2] Page 10 line 239 'unsteady aerodynamics model' -> unsteady airfoil aerodynamics model'*

   **[Authors]** We implemented the change as suggested.

33. *[Reviewer #2] Page 13 line 285 'influnece' -> influence*

   **[Authors]** We corrected this typo.

34. *[Reviewer #2] Page 23 line 400 'the normalised overshoot for the Mflap and Fthrust is similar per pitch direction, whereas the overshoot in Maero is 3 to 4.5 times higher'. I believe this is because the dynamic inflow effect causes the inflow angle and angle of attack to lag behind the quasi steady value. The thrust and the flapwise moment feel the effect mainly due to a change in magnitude of the lift force (due to the lag of the angle of attack), while the torque feels this change in magnitude and also the change in the projection of the lift force in the in-plane direction due to the lag of the inflow angle (Equation 12). Because the thrust force is determined using the cosine of the inflow angle (Equation 11), the effect of the inflow angle lag is much smaller there.*

   **[Authors]** Thank you for this excellent comment. We have added this argumentation closely based on your text in the discussion of the load overshoot (line 596-601)

   > The higher relative overshoot of the torque by a factor of 3 to 4.5 compared to the flapwise blade root bending moment and the rotor thrust can be related to the lag in inflow angle and thus angle of attack behind the quasi steady value. $M_{\text{flap}}$ and $F_{\text{thrust}}$ feel the effect mainly due to the change in the magnitude of the lift force, while $M_{\text{aero}}$ feels the change in magnitude

   > and also the change of the projection of the lift force in the tangential direction due to the lag of the inflow angle (see Eq. 12).
   > 600 Because the $F_{\text{thrust}}$ and mostly also $M_{\text{flap}}$ are determined by the cosine of the inflow angle (see Eq. 11), the effect of the inflow angle lag is much smaller here.

35. *[Reviewer #2] Page 28 line 563 'by the shed vortex from the tip due to the fast change in trailed vorticity'. In the literature, sometimes 'shed vorticity' is used to describe vorticity that is parallel to the span, and 'trailed vorticity' to describe vorticity that is perpendicular to the span. Maybe you could instead write 'by the tip vortex due to the fast change in trailed vorticity'*

   **[Authors]** Thank your for making us aware of this unclear description. We have changed the sentence as proposed (line 638).

[revised manuscript text omitted]